# ER-luminal [Ca$^{2+}$] regulation of InsP$_3$ receptor gating mediated by an ER-luminal peripheral Ca$^{2+}$-binding protein

Horia Vais[1], Min Wang[1], Karthik Mallilankaraman[1], Riley Payne[1], Chris McKennan[2], Jeffrey T Lock[3], Lynn A Spruce[4], Carly Fiest[1], Matthew Yan-lok Chan[1], Ian Parker[3,5], Steven H Seeholzer[4], J Kevin Foskett[1,6]*, Don-On Daniel Mak[1]*

[1]Department of Physiology, Perelman School of Medicine, University of Pennsylvania, Philadelphia, United States; [2]Department of Statistics, University of Pittsburgh, Pittsburgh, United States; [3]Department of Neurobiology and Behavior, University of California, Irvine, United States; [4]Proteomics Core Facility, The Children's Hospital of Philadelphia, Philadelphia, United States; [5]Department of Physiology and Biophysics, University of California, Irvine, United States; [6]Department of Cell and Developmental Biology, Perelman School of Medicine, University of Pennsylvania, Philadelphia, United States

*For correspondence:
foskett@pennmedicine.upenn.edu
(JKF);
dodmak@gmail.com (D-ODM)

Competing interests: The authors declare that no competing interests exist.

**Abstract** Modulating cytoplasmic Ca$^{2+}$ concentration ([Ca$^{2+}$]$_i$) by endoplasmic reticulum (ER)-localized inositol 1,4,5-trisphosphate receptor (InsP$_3$R) Ca$^{2+}$-release channels is a universal signaling pathway that regulates numerous cell-physiological processes. Whereas much is known regarding regulation of InsP$_3$R activity by cytoplasmic ligands and processes, its regulation by ER-luminal Ca$^{2+}$ concentration ([Ca$^{2+}$]$_{ER}$) is poorly understood and controversial. We discovered that the InsP$_3$R is regulated by a peripheral membrane-associated ER-luminal protein that strongly inhibits the channel in the presence of high, physiological [Ca$^{2+}$]$_{ER}$. The widely-expressed Ca$^{2+}$-binding protein annexin A1 (ANXA1) is present in the nuclear envelope lumen and, through interaction with a luminal region of the channel, can modify high-[Ca$^{2+}$]$_{ER}$ inhibition of InsP$_3$R activity. Genetic knockdown of ANXA1 expression enhanced global and local elementary InsP$_3$-mediated Ca$^{2+}$ signaling events. Thus, [Ca$^{2+}$]$_{ER}$ is a major regulator of InsP$_3$R channel activity and InsP$_3$R-mediated [Ca$^{2+}$]$_i$ signaling in cells by controlling an interaction of the channel with a peripheral membrane-associated Ca$^{2+}$-binding protein, likely ANXA1.

## Introduction

Modulating cytoplasmic free Ca$^{2+}$ concentration ([Ca$^{2+}$]$_i$) is a universal signaling pathway that regulates numerous cell-physiological processes (*Berridge, 2016*). Ubiquitous endoplasmic reticulum (ER)-localized inositol 1,4,5-trisphosphate (InsP$_3$) receptor (InsP$_3$R) Ca$^{2+}$-release channels play a central role in this pathway (*Foskett et al., 2007*). InsP$_3$ generated in response to extracellular stimuli binds to and activates InsP$_3$R channels to release Ca$^{2+}$ stored in the ER lumen, generating diverse local and global [Ca$^{2+}$]$_i$ signals (*Berridge, 2016*). Whereas much is known regarding regulation of InsP$_3$R channel gating by multiple processes, including binding of cytoplasmic ligands (Ca$^{2+}$, InsP$_3$ and ATP$^{4-}$), post-translational modifications, interactions with proteins, clustering and differential localization (*Berridge, 2016*), the regulation of InsP$_3$R channel activity by Ca$^{2+}$ concentration in the ER lumen ([Ca$^{2+}$]$_{ER}$) is poorly understood and controversial (see (*Caroppo et al., 2003*; *Yamasaki-Mann and Parker, 2011*) and references therein). InsP$_3$R activity influences [Ca$^{2+}$]$_{ER}$, which is critical

for many processes, including regulation of bioenergetics, protein biogenesis and folding, and $Ca^{2+}$ signaling (*Corbett and Michalak, 2000*). $[Ca^{2+}]_{ER}$ dysregulation is associated with pathological conditions, including ER stress responses, diabetes, cardiac dysfunction, neurodegeneration, defective cell proliferation and cell death (*Berridge, 2016*). Mechanisms that link cellular processes with $[Ca^{2+}]_{ER}$, and the connections between $[Ca^{2+}]_{ER}$ dysregulation and disease pathogenesis are ill-defined and poorly understood (*Mekahli et al., 2011*).

For a long time, regulation of $InsP_3R$ channel activity by $[Ca^{2+}]_{ER}$ has primarily been examined by approaches that relied on changes in $[Ca^{2+}]_i$ or $[Ca^{2+}]_{ER}$ to infer channel activity (*Caroppo et al., 2003*; *Yamasaki-Mann and Parker, 2011*) and references therein), with no rigorous control of both $[Ca^{2+}]_{ER}$ and $[Ca^{2+}]_i$ simultaneously. It has therefore been difficult to differentiate feed-through effects in which $Ca^{2+}$ flux through channel raises $[Ca^{2+}]_i$ at the cytoplasmic $Ca^{2+}$-binding sites to modulate channel activity from direct effects of $[Ca^{2+}]_{ER}$ on the luminal aspect of the $InsP_3R$. Patch-clamp electrophysiology allows rigorous, simultaneous control of $[Ca^{2+}]$ on both sides of the membrane, but intracellular ER membranes are not accessible to patch pipettes. To overcome this limitation, we made use of the fact that the ER membrane is continuous with the outer nuclear membrane (*Dingwall and Laskey, 1992*) and pioneered the use of nuclear patch-clamp electrophysiology on isolated nuclei (*Mak et al., 2005*). This enables the activities of single $InsP_3R$ channels to be recorded in their native ER membrane and luminal milieus. Different patch-clamp configurations have been used to study single $InsP_3R$ channels under rigorously controlled $[Ca^{2+}]_{ER}$ *and* $[Ca^{2+}]_i$: including on-nucleus (on-nuc, cytoplasmic aspect of the channel faces into the pipette solution with the luminal milieu intact, *Figure 1A*), excised luminal-side-out (lum-out, cytoplasmic aspect of the channel faces into the pipette solution with the luminal aspect exposed to the bath solution, *Figure 1B*) and excised cytoplasmic-side-out (cyto-out, cytoplasmic aspect of the channel perfused by the bath solution with the luminal aspect facing the pipette solution, *Figure 1C*; *Mak et al., 2013a*; *Mak et al., 2013b*). Because of the relatively low selectivity of $InsP_3R$ channels for $Ca^{2+}$ vs $K^+$ (15.2 : 1 [*Vais et al., 2010a*]) and orders of magnitude higher $[K^+]$ (140 mM) than that of other ions in the physiological solutions used in our experiments (70 nM to $\leq$600 μM $[Ca^{2+}]_{free}$; 0 or 200 μM $[Mg^{2+}]_{free}$), the electrical currents through open $InsP_3R$ channels are overwhelmingly carried by $K^+$ in all our patch-clamp electrophysiology experiments, enabling the kinetics of channel gating to be studied with both luminal and cytoplasmic $[Ca^{2+}]$ well-defined and controlled.

Using the nuclear patch-clamp approach in a previous study (*Vais et al., 2012*), we demonstrated that $InsP_3R$ channel activity can be modulated by $[Ca^{2+}]_{ER}$ indirectly via feed-through effects of $Ca^{2+}$ flux driven through an open channel by high $[Ca^{2+}]_{ER}$ that raises the local $[Ca^{2+}]_i$ in the channel vicinity to regulate its activity through its cytoplasmic activating and inhibitory $Ca^{2+}$-binding sites (*Figure 2A*). That study demonstrated that these feed-through effects can be completely abrogated by sufficient $Ca^{2+}$ chelation on the cytoplasmic side to buffer local $[Ca^{2+}]_i$ at cytoplasmic $Ca^{2+}$

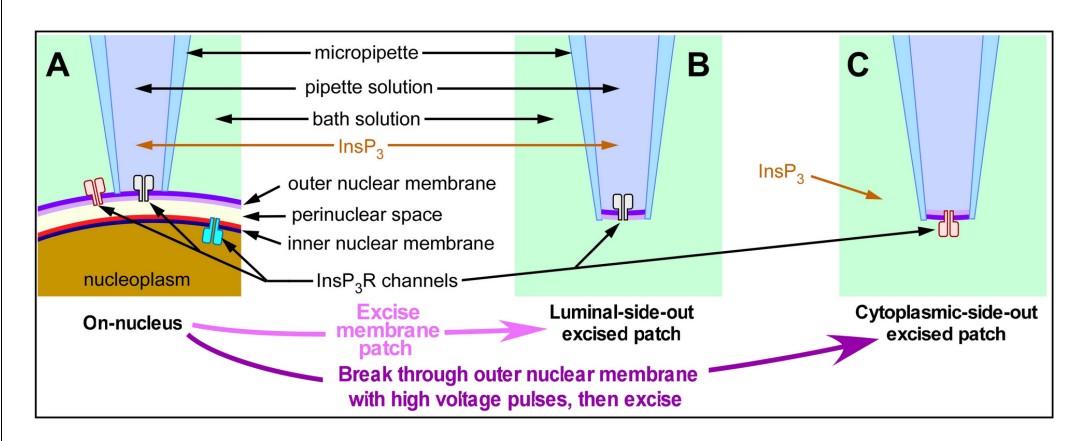

**Figure 1.** Schematic diagram illustrating the orientation of $InsP_3R$ channels in isolated nuclear membrane patches and $InsP_3$-containing solution relative to the micropipette in various configurations of nuclear patch-clamping. (**A**) On-nucleus configuration with outer nuclear membrane intact, (**B**) excised luminal-side-out configuration, (**C**) excised cytoplasmic-side-out configuration.

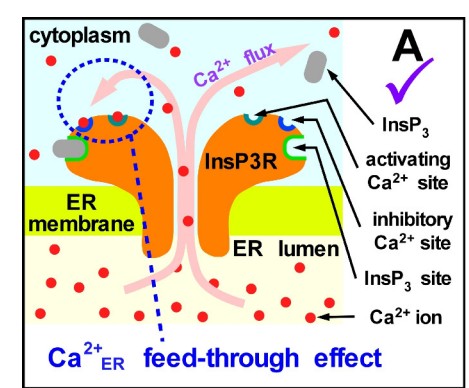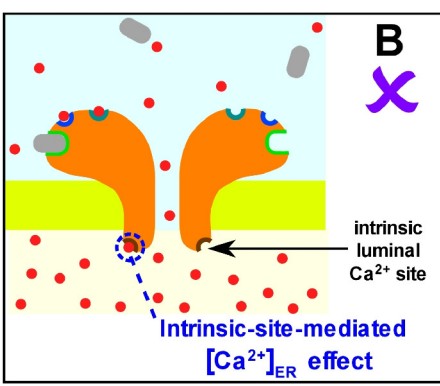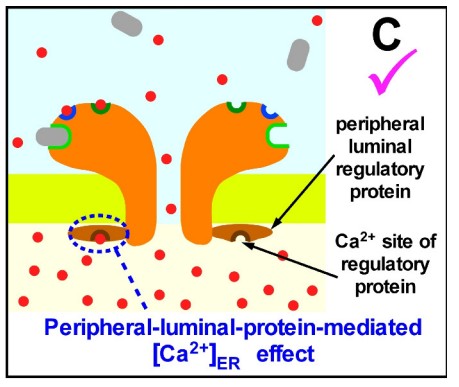

**Figure 2.** Possible mechanisms of $[Ca^{2+}]_{ER}$ regulation of $InsP_3R$ channel activity. (**A**) $Ca^{2+}$ flux through an open $InsP_3R$ channel driven by high $[Ca^{2+}]_{ER}$ raises local $[Ca^{2+}]_i$ in the pore vicinity to regulate the channel through its cytoplasmic activating and inhibitory $Ca^{2+}$-binding sites. (**B**) $Ca^{2+}$ binds directly to an intrinsic site on the luminal side of the channel to regulate its activity. (**C**) $Ca^{2+}$ binding to a peripheral protein in the ER lumen regulates channel activity indirectly, by promoting interaction of the peripheral protein with the $InsP_3R$.

binding sites of the $InsP_3R$. In the presence of 5 mM 5,5'-diBromo BAPTA (a fast acting $Ca^{2+}$ chelator) on the cytoplasmic side in the lum-out excised patch configuration (*Figure 1B*), the open probability $P_o$ of the $InsP_3R$ channel did not change discernably when the $[Ca^{2+}]_{ER}$ was switched between 70 nM and 300 µM, no matter whether saturating 10 µM $[InsP_3]$ (Figure 2C in *Vais et al., 2012*) or sub-saturating 3 µM $[InsP_3]$ (Figure 7B in *Vais et al., 2012*) was present in the micropipette (2 µM $[Ca^{2+}]_i$ was in the micropipette in all experiments described here). Thus, as long as $[Ca^{2+}]_{free}$ on the cytoplasmic side was well buffered, switching $[Ca^{2+}]_{ER}$ between 70 nM and 300 µM in lum-out excised membrane patches had no effect on channel activity. This definitively ruled out the possibility that $InsP_3R$ activity is modulated directly by an intrinsic $Ca^{2+}$-regulatory site on the luminal side that is either part of the $InsP_3R$ or of a protein that constitutively interacts with the channel (*Figure 2B*).

Notably, however, in all previous studies of $InsP_3R$ channel gating, nuclei were isolated into a bath solution with low $[Ca^{2+}]_{free}$ to simulate physiological resting $[Ca^{2+}]_i$ (*Mak and Foskett, 2015*). Although sarco/endoplasmic reticulum $Ca^{2+}$-ATPase (SERCA) is present in the nuclear envelope (*Lanini et al., 1992*), the bath solutions contained no MgATP to energize it. Thus, passive $Ca^{2+}$ leaks equilibrated $[Ca^{2+}]_{free}$ in the perinuclear space of the isolated nuclei (effectively $[Ca^{2+}]_{ER}$) with bath $[Ca^{2+}]$ at unphysiologically low levels. Accordingly, possible $[Ca^{2+}]_{ER}$ regulation of channel activity by peripheral proteins in the lumen that interact with the $InsP_3R$ only in the presence of high, physiological $[Ca^{2+}]_{ER}$ (*Figure 2C*) would have been undetected in previous studies.

Here we specifically assessed whether such a peripheral luminal protein (PLP)-mediated $[Ca^{2+}]_{ER}$ regulation of $InsP_3R$ channel activity exists. We discovered a strong $Ca^{2+}$ flux-independent inhibition of $InsP_3R$ channel activity mediated by accessory ER-luminal protein(s) only in the presence of high, physiological $[Ca^{2+}]_{ER}$. We identified the region of the $InsP_3R$ that is involved in this $[Ca^{2+}]_{ER}$-dependent inhibition, and discovered that the widely expressed $Ca^{2+}$-binding protein annexin A1 (ANXA1) localizes not only to the cytoplasm but also to the perinuclear space inside the nuclear envelope, and possibly in the ER lumen as well, where it plays a critical role in channel inhibition through high $[Ca^{2+}]_{ER}$-mediated interaction with the $InsP_3R$. Reducing this interaction enhanced agonist-induced $InsP_3R$-mediated $Ca^{2+}$ release, and increased local elementary $Ca^{2+}$-release events in intact cells. Thus, $[Ca^{2+}]_{ER}$ is a major regulator of $InsP_3R$-channel activity and $InsP_3R$-mediated $[Ca^{2+}]_i$ signaling in cells by controlling an interaction of the channel with a luminal peripheral membrane-associated $Ca^{2+}$-binding protein, likely ANXA1.

## Results

### Novel regulation by [Ca$^{2+}$]$_{ER}$ of InsP$_3$R single-channel activity

We performed single-channel patch-clamp electrophysiology on outer membranes of nuclei isolated from mutant chicken B cells in which all three endogenous InsP$_3$R genes were ablated and stably replaced with wild-type rat type-3 InsP$_3$R (DT40-r3 cells) (*Mak et al., 2005*), so that homotetrameric rat type-3 InsP$_3$R channels (*Vais et al., 2012*) were recorded in our experiments. In on-nuc patch-clamp experiments with the bath containing no MgATP, the [Ca$^{2+}$]$_{ER}$ in the perinuclear space (*Figure 3A*) equilibrated with [Ca$^{2+}$]$_{bath}$ (70 nM). Linear, ohmic InsP$_3$R-channel current vs. applied potential ($i_{ch}$-$V_{app}$) curves were observed with high single-channel conductance, as expected (*Figure 3B*). In contrast, with 1 mM MgATP in the bath solution (*Figure 3C*), SERCA activity raised [Ca$^{2+}$]$_{ER}$ to physiological levels, if the outer and inner membranes of the isolated nuclei remained intact. The $i_{ch}$-$V_{app}$ curves (*Figure 3D*) observed were non-ohmic and asymmetric with respect to the origin, and single-channel conductance was reduced, both caused by permeant-ion (Ca$^{2+}$) block of the measured K$^+$ current due to high [Ca$^{2+}$]$_{ER}$. Thus, absence or presence of MgATP in the bath provides a mechanism to alter [Ca$^{2+}$]$_{ER}$ in intact nuclei. In a previous study examining the Ca$^{2+}$ permeant-ion effect on K$^+$ conductance through InsP$_3$R channels (*Vais et al., 2010a*), we derived an empirical equation that describes, with high accuracy, the InsP$_3$R channel slope conductance over a broad range of [Ca$^{2+}$]$_{ER}$ (0 to 1.1 mM). This enabled us to use the single-channel conductance, rather than Ca$^{2+}$ indicator dye fluorescence, to estimate that [Ca$^{2+}$]$_{ER}$ is ~300 µM in the perinuclear space of intact isolated DT40-r3 nuclei in the presence of bath MgATP. Subsequent excision of the membrane patch into the lum-out patch configuration (*Mak et al., 2013c*) exposed its luminal side to 70 nM [Ca$^{2+}$]$_{bath}$ (*Figure 3E*), which restored linear $I_{ch}$-$V_{app}$ curves and high single-channel conductance (*Figure 3F*).

In on-nuc patches of DT40-r3 nuclei, InsP$_3$R channel open probability ($P_o$) was high with 70 nM Ca$^{2+}$$_{ER}$ (*Figure 3H and L*, red) whereas it was profoundly reduced with [Ca$^{2+}$]$_{ER}$ raised to 300 µM by addition of MgATP to the bath (*Figure 3H and L*, blue). Even in saturating (10 µM) [InsP$_3$], 300 µM Ca$^{2+}$$_{ER}$ strongly reduced $P_o$ (*Figure 3I and M*, blue), to a level indistinguishable from that observed in 3 µM InsP$_3$ (*Figure 3L*, blue). In contrast, when [Ca$^{2+}$]$_{ER}$ = 70 nM, $P_o$ was higher in 10 µM InsP$_3$ (*Figure 3M*, red) than in 3 µM InsP$_3$ (*Figure 3L*, red). With the bath solution containing 1 mM MgATP and the SERCA inhibitor thapsigargin (2.5 µM) (*Figure 3M*, purple and *Figure 3—figure supplement 1*), $P_o$ was indistinguishable from that in 0-MgATP$_{bath}$, confirming that bath MgATP raised [Ca$^{2+}$]$_{ER}$ in intact nuclei by supporting SERCA activity. Because high concentrations of Ca$^{2+}$ chelator (5 mM diBrBAPTA) buffered [Ca$^{2+}$]$_i$ at cytoplasmic Ca$^{2+}$ regulatory sites, possible feed-through effects were eliminated (*Vais et al., 2012*). Thus, with an intact ER-luminal milieu and [Ca$^{2+}$]$_{ER}$ at a level characteristic of replete ER stores, InsP$_3$R activity is profoundly suppressed by a mechanism unrelated to but as powerful as the well-known high-[Ca$^{2+}$]$_i$ inhibition of InsP$_3$R activity. Because luminal [Ca$^{2+}$] does not directly affect InsP$_3$R channel activity (*Vais et al., 2012*), these results suggest a novel regulatory mechanism, possibly mediated by a resident luminal Ca$^{2+}$-binding protein.

After recording channel activity in the on-nuc patch-clamp configuration, patches were excised into the lum-out configuration to observe gating of channels with their luminal sides exposed to bath [Ca$^{2+}$]$_{free}$ = 70 nM. Since [Ca$^{2+}$]$_i$ and [Ca$^{2+}$]$_{ER}$ experienced by channels in this configuration were effectively the same as those in the on-nuc configuration with no bath MgATP, it was expected that the inhibitory luminal-protein-mediated effect would be abrogated. In agreement, similar $P_o$ were observed for channels in this lum-out configuration (*Figure 3J and M*, green) and channels in the on-nuc configuration in 0-MgATP (*Figure 3I and M*, red). Importantly, channel $P_o$ remained high during subsequent perfusion with 300 µM Ca$^{2+}$$_{free}$ (*Figure 3K and M*, brown). Lack of inhibition when [Ca$^{2+}$] was raised to 300 µM suggests that the luminal effector mediating high-[Ca$^{2+}$]$_{ER}$ inhibition is only loosely associated with the InsP$_3$R, becoming irretrievably lost when the luminal side of the isolated patch was perfused by low-[Ca$^{2+}$]$_{free}$ solution.

Since the levels of suppression of InsP$_3$R channel activities in the presence of high [Ca$^{2+}$]$_{ER}$ were independent of [InsP$_3$] used (*Figure 3L and M*), all subsequent experiments were performed with saturating 10 µM [InsP$_3$] to avoid possible ambiguous results due to insufficient stimulation of InsP$_3$R channels.

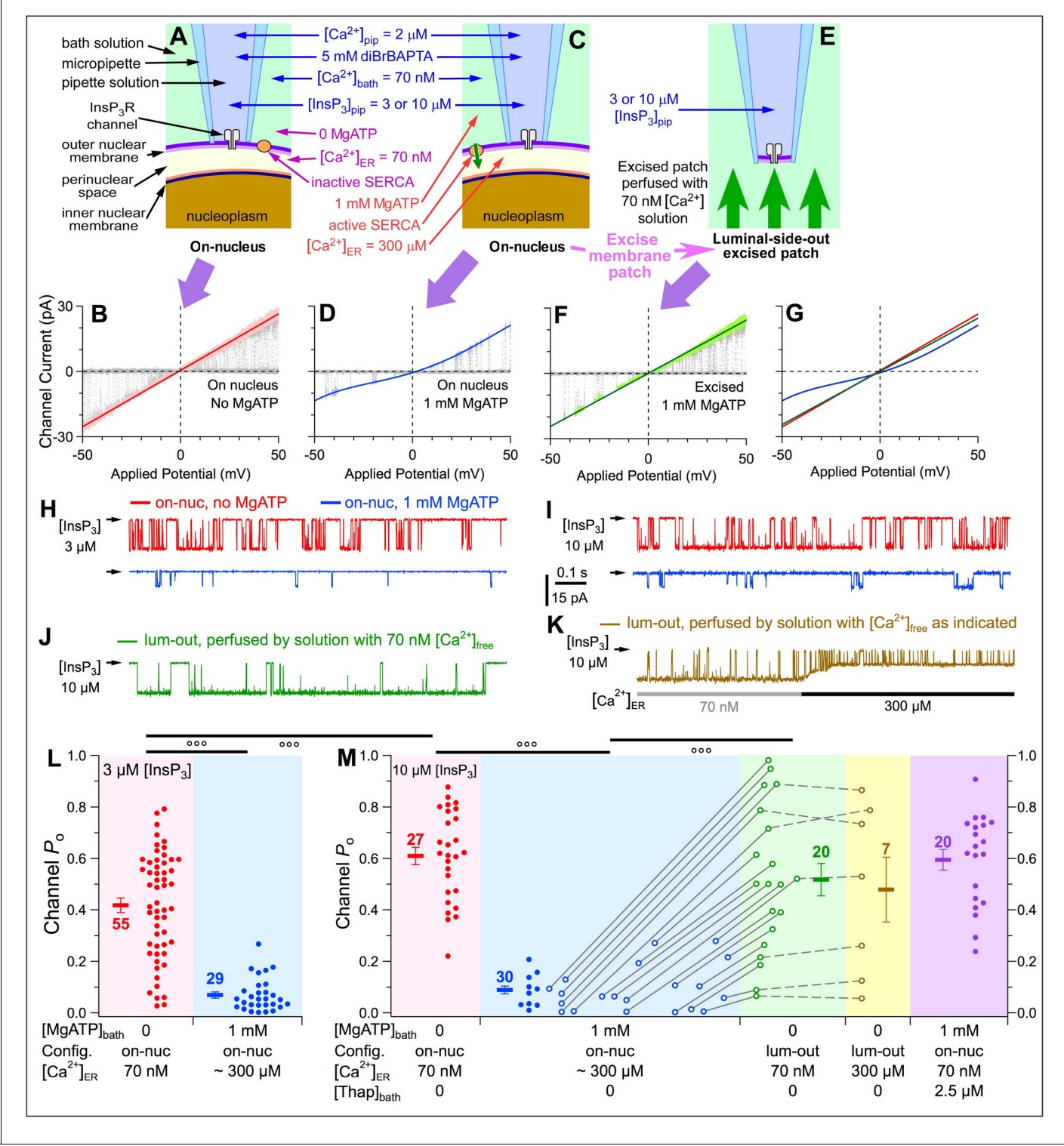

**Figure 3.** Raising $[Ca^{2+}]_{ER}$ in intact isolated nuclei affects activities of $InsP_3R$ channels. (**A and B**) On-nucleus (on-nuc) patch-clamp configuration and single-channel current-voltage ($i_{ch}$-$V_{app}$) plot of $InsP_3R$ channels in bath solution with 0-MgATP so SERCA was not active and $[Ca^{2+}]_{ER}$ equilibrated with $[Ca^{2+}]_{bath}$. (**C and D**) On-nuc configuration and $i_{ch}$-$V_{app}$ plot of $InsP_3R$ channels in bath solution with 1 mM MgATP to activate SERCA to move $Ca^{2+}$ from the bath into the perinuclear space, raising $[Ca^{2+}]_{ER}$ to ~300 µM. ($[Ca^{2+}]_{ER}$ was estimated from the magnitude of the $InsP_3R$ channel current size [**Vais et al., 2010a**]). (**E and F**) Excised luminal-side-out (lum-out) configuration and $i_{ch}$-$V_{app}$ plot of $InsP_3R$ channels in isolated membrane patch perfused with 70 nM $[Ca^{2+}]_{ER}$ and 0-MgATP bath solution. The baseline closed-channel currents were subtracted from the $i_{ch}$-$V_{app}$ plots. Cytoplasmic $[InsP_3]$ in the pipette solution = 3 µM for sub-saturating level, and 10 µM for saturating level. $V_{app}$ ramped from –50 to 50 mV (w.r.t. ground electrode in

*Figure 3 continued on next page*

*Figure 3 continued*

the bath) in 2 s. 10–20 ramps were averaged for each graph. Colored lines: linear (**B, F**) and 4$^{th}$ order polynomial (**D**) fits to open-channel data points in graphs. In this and all subsequent patch-clamp experiments, optimal cytoplasmic [Ca$^{2+}$]$_{free}$ (2 µM) was used. In experiments shown here, [Ca$^{2+}$]$_{free}$ in bath solution = 70 nM. (**G**) Overlay of three fitted curves in (**B, D and F**). (**H**) Typical single-channel current traces recorded under constant $V_{app}$ (–30 mV) in on-nuc configuration, with pipette solution containing sub-saturating 3 µM InsP$_3$, and bath solutions containing 0- (red) or 1 mM-MgATP (blue). In this and all subsequent current traces, arrow on left of trace indicates closed-channel current level. (**I**) Corresponding current traces recorded with pipette solution containing saturating 10 µM InsP$_3$. (**J**) Typical current trace recorded in lum-out configuration with constant $V_{app}$ = –30 mV. (**K**) Typical current trace recorded as lum-out patch was perfused by solution with 70 nM Ca$^{2+}$$_{free}$ (grey bar) and then switched to one with 300 µM Ca$^{2+}$$_{free}$ (black bar). (**L**) $P_o$ from individual current traces (filled circles), and their averages (thick horizontal bars) and s.e.m. (error bars) observed in on-nuc patch-clamp configuration with 3 µM InsP$_3$ in pipette solution, and with 0- (red) or 1 mM (blue) MgATP in bath solutions. [Ca$^{2+}$]$_{free}$ on luminal side tabulated at x-axis. Numbers of current traces tabulated next to corresponding averages. In this and all subsequent data plots, symbols °, °° and °°° indicate t-test p value < 0.05, 0.005 and 0.001, respectively. (**M**) $P_o$ from individual current traces (circles), averages (thick bars) and s.e.m. (error bars) observed with 10 µM InsP$_3$ in pipette solution. Red symbols: $P_o$ in on-nuc configuration with 0-MgATP in bath so [Ca$^{2+}$]$_{ER}$ = [Ca$^{2+}$]$_{bath}$ = 70 nM. Blue symbols: $P_o$ in on-nuc configuration with 1 mM MgATP in bath so [Ca$^{2+}$]$_{ER}$ ~300 µM. Filled circles: $P_o$ from experiments in which only the on-nuc configuration was achieved. Open circles: $P_o$ from experiments in which lum-out configuration was achieved after on-nuc channel activity had been recorded. Green symbols: $P_o$ observed in lum-out patches whose luminal side was perfused with solution containing 70 nM Ca$^{2+}$$_{free}$. Open circles connected with grey lines: $P_o$ observed in same patch before and after membrane excision. Brown symbols: $P_o$ observed in lum-out membrane patches after switching to perfusing solution containing 300 µM Ca$^{2+}$$_{free}$, as in (**K**). Open circles connected with dashed grey lines: $P_o$ observed in same lum-out membrane patch before and after perfusion-solution switching. Purple symbols: $P_o$ in on-nuc configuration with 1 mM bath MgATP and 2.5 µM thapsigargin.

The online version of this article includes the following figure supplement(s) for figure 3:

**Figure supplement 1.** Thapsigargin abrogated SERCA activity despite the presence of bath MgATP.

To determine whether peripheral luminal-protein (PLP) inhibition affects endogenous InsP$_3$R in different cell types, similar on-nuc patch-clamp experiments were conducted with intact nuclei (verified by asymmetric $i_{ch}$-$V_{app}$ curves [*Figure 4A–C*] and reduced single-channel conductance [*Figure 4D–F*]) isolated from mouse N2a cells, which predominantly express type 1 InsP$_3$R (InsP$_3$R–1) (*Wojcikiewicz, 1995*), rat PC-12 cells expressing mainly InsP$_3$R-1 and InsP$_3$R-3 (*Newton et al., 1994*), and wild-type (WT) chicken DT40 cells, which express all three isoforms (*Sugawara et al., 1997*). In a bath containing 0-MgATP, channels from the different cell types exhibited different maximum $P_o$ (*Figure 4D–F* red traces, and *Figure 4G*). With high [Ca$^{2+}$]$_{ER}$ generated by 1 mM bath MgATP, $P_o$ of the various endogenous InsP$_3$R channels were strongly suppressed (*Figure 4D–F* blue traces, and *Figure 4H*). Interestingly, $P_o$ of the different InsP$_3$Rs were strongly suppressed to very similar extents ($P_o$ reduction: WT DT40 92%, PC-12 86%, N2a 90% (*Figure 4I*), comparable to that of InsP$_3$R-3 in DT40-r3 cells (85%) (*Figure 3M*, red and blue). This suggests that the inhibitory protein is present in the ER lumen of many cell types and interacts with all three InsP$_3$R isoforms under high [Ca$^{2+}$]$_{ER}$.

## A luminal loop of the InsP$_3$R is critical for luminal protein-mediated [Ca$^{2+}$]$_{ER}$ regulation

We hypothesized that the PLP interacts directly with the InsP$_3$R. Only a small fraction of the InsP$_3$R sequence is exposed to the ER lumen, consisting of three loops (L1, L2, and L3) connecting trans-membrane helices in the pore-forming domain. We selected human sequences (*Figure 5A–C*) based on structural information about the InsP$_3$R (*Fan et al., 2015*). Since the inhibitory effects were observed in cells expressing different channel isoforms, we selected sequences conserved in all three InsP$_3$R isoforms. We excluded the P-region (orange stripe) and selectivity filter (green stripe) involved in other crucial channel functions (*Figure 5C*). Only four short sequences (magenta stripes in *Figure 5B–C*) fit the criteria to be luminal region(s) of the InsP$_3$R that could interact with the PLP.

We first investigated whether the conserved region of the L2 loop (pL2, *Figure 5B*) is involved, by recording InsP$_3$R-3 channels in the cytoplasmic-side-out (cyto-out) configuration (*Mak et al., 2013b*) with pipette solutions that faced the luminal aspect of the channel containing synthetic pL2 peptides. Nuclei were first exposed to [Ca$^{2+}$]$_{bath}$ = 70 nM and 0-MgATP, so [Ca$^{2+}$]$_{ER}$ equilibrated with [Ca$^{2+}$]$_{bath}$ at 70 nM (*Figure 5D*, left) promoting dissociation of PLP from the InsP$_3$R. When the cyto-out patch configuration was achieved (*Figure 5D*, right), the luminal aspect of the channel faced the pipette solution containing [Ca$^{2+}$]$_{pip}$ = 300 µM. Because the PLP had already dissociated from the channel before patch excision (*Figure 5D*, left), we expected no inhibition to be observed.

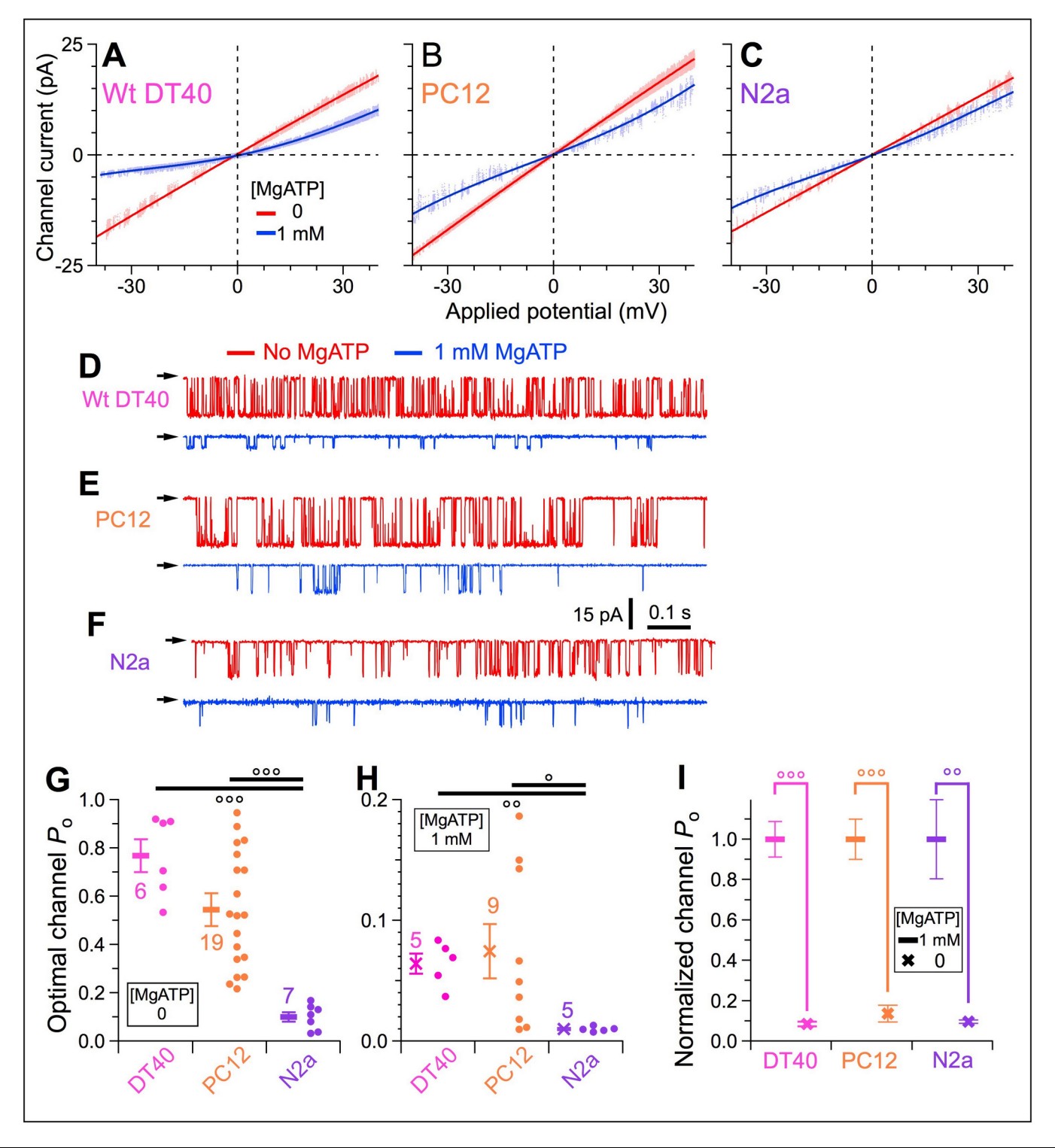

**Figure 4.** Activities of InsP$_3$R channels from a number of cell lines expressing various InsP$_3$R isoforms in different proportions are similarly affected by rise in [Ca$^{2+}$]$_{ER}$ in intact isolated nuclei. (A–C) Single-channel $I$-$V_{app}$ plots of on-nuc patch-clamp experiments of endogenous InsP$_3$R channels from WT DT40 (A), PC12 (B), and N2a (C) cells. Channels activated by 10 μM InsP$_3$ with (blue) or without (red) 1 mM MgATP. Data obtained using protocol as in *Figure 3* with $V_{app}$ ramped from –40 to 40 mV in 2 s. Lines: linear (red) and 4$^{th}$ order polynomial (blue) fits to open-channel data points. (D–F) Typical current traces for endogenous InsP$_3$R channels from WT DT40 (D), PC12 (E), and N2a (F) cells, in presence (blue) and absence (red) of 1 mM MgATP in bath with $V_{app}$ = –40 mV. (G) Optimal $P_o$ activated by 10 μM InsP$_3$ and 2 μM Ca$^{2+}_i$ in individual experiments (circles) and averages and s.e.m. (bars) for

*Figure 4 continued on next page*

Figure 4 continued

endogenous WT DT40 (magenta), PC12 (orange) and N2a (purple) InsP$_3$R channels in bath without MgATP. (**H**) Optimal $P_o$ in bath with 1 mM bath MgATP activated by the same ligand conditions as (**G**) for the same endogenous InsP$_3$R channels. (**I**) Normalized $P_o$, their averages and s.e.m. for endogenous InsP$_3$R channels in bath with 1 mM MgATP, relative to respective average optimal $P_o$ in (**G**).

Indeed, when the patch was perfused with saturating 10 µM InsP$_3$ and optimal 2 µM Ca$^{2+}$$_i$ (buffered with 5 mM dibromoBAPTA) (**Vais et al., 2010b**), channels were activated with reduced conductance due to permeant Ca$^{2+}$ block caused by high [Ca$^{2+}$]$_{ER}$ in the pipette solution, but they exhibited high $P_o$ (~0.6) (**Figure 5G and J**, magenta) indistinguishable from that observed in on-nuc experiments under similar conditions (red bars in **Figure 5J**). Alternately, nuclei were first exposed to 1 mM MgATP in the bath to raise [Ca$^{2+}$]$_{ER}$ to ~300 µM to promote an interaction between the PLP and the InsP$_3$R (**Figure 5E**, left). Because the pipette solution contained 300 µM Ca$^{2+}$, the interaction remained intact when the cyto-out patch configuration was achieved (**Figure 5E**, right). Indeed, low channel $P_o$ was observed with maximal stimulation (**Figure 5H and J**, purple), similar to the observations made using the on-nuc configuration (**Figure 5J**, blue bars). Having therefore established that the inhibitory effect of the PLP could be observed in the cyto-out patch configuration, we included 10 µM pL2 peptide in the pipette solution (**Figure 5F**, left). With pL2 peptides present on the cytoplasmic side in the on-nuc configuration before establishing the cyto-out configuration, InsP$_3$R remained bound to the PLP before membrane excision. However, after achieving the cyto-out configuration, the luminal side of the InsP$_3$R became exposed to the peptides. We reasoned that if the pL2 region is involved in the inhibitory effects through interactions with the PLP, then pL2 peptide in high concentrations should compete the PLP away from the InsP$_3$R, and thereby raise $P_o$ despite presence of 300 µM Ca$^{2+}$ (**Figure 5F**, right). Indeed, inhibition was completely abolished such that the $P_o$ of channels with pL2 peptides on their luminal side (**Figure 5I–J**, brown) was indistinguishable from $P_o$ recorded in on-nuc (red) or cyto-out patch-clamp experiments with 0-MgATP in the bath (magenta). This indicates that the pL2 region interacts with the PLP to suppress InsP$_3$R channel activity, and that the inhibitory effect of the PLP is mostly or completely mediated by the pL2-PLP interaction.

## Identification of the inhibitory peripheral protein(s) in the ER lumen

To identify the PLP, peptides with a modified pL2 sequence (mpL2) or scrambled mpL2 (smpL2) were used for in-vitro pull-downs from soluble fractions of bovine hepatocyte ER microsomes (**Figure 6—figure supplement 1**). Eluates from the pull-downs were analyzed by mass spectrometry. Among >460 proteins detected, we looked for those enriched in the mpL2, that are Ca$^{2+}$-binding proteins (UniProt protein database) in the ER lumen (or in topologically-equivalent locations), and that are broadly expressed. The annexin (ANX) protein family met these criteria. Of 12 members of the ANX family, five were identified to have statistically-significant preference to bind to the mpL2 peptide over the smpL2 peptide (**Figure 6—source data 1**, also see 'Ca$^{2+}$-dependent affinity enrichment mass spectrometry' in Materials and methods).

ANX proteins are ubiquitously expressed, Ca$^{2+}$-dependent phospholipid-binding proteins (**Raynal and Pollard, 1994**). Ca$^{2+}$ binding to ANX (dissociation constant $K_d$ ~25 µM to 1 mM) promotes its association with negatively-charged phospholipid-containing membranes (**Raynal and Pollard, 1994**; **Gerke and Moss, 2002**). Although ANX proteins are known to be cytoplasmic, many studies have reported their presence in locations topologically equivalent to the ER lumen, including the extracellular side of the plasma membrane (PM) (**Solito et al., 1994**; **Brownstein et al., 2001**; **McArthur et al., 2009**; **Yáñez et al., 2012**; **Mirkowska et al., 2013**), within intracellular and secretory granules (**McArthur et al., 2009**; **Leoni et al., 2015**; **Chi et al., 2006**; **Boudhraa et al., 2016**; **Perretti et al., 2000**), and in the extracellular space (**McArthur et al., 2009**; **Boudhraa et al., 2016**; **Arur et al., 2003**; **Fan et al., 2004**; **Scannell et al., 2007**; **D'Acquisto et al., 2007**; **Iwasa et al., 2012**; **Belvedere et al., 2014**), suggesting that ANX can also be present in the ER lumen.

Among annexins, ANXA1 is most widely reported to be present in locations topologically equivalent to the ER lumen. To confirm that ANXA1 was present in the ER lumen, we localized it by immunofluorescence microscopy. A549 cells were briefly exposed to a low concentration of digitonin to permeabilize the PM to release cytoplasmic ANXA1. Remaining ANXA1 co-localized with the ER

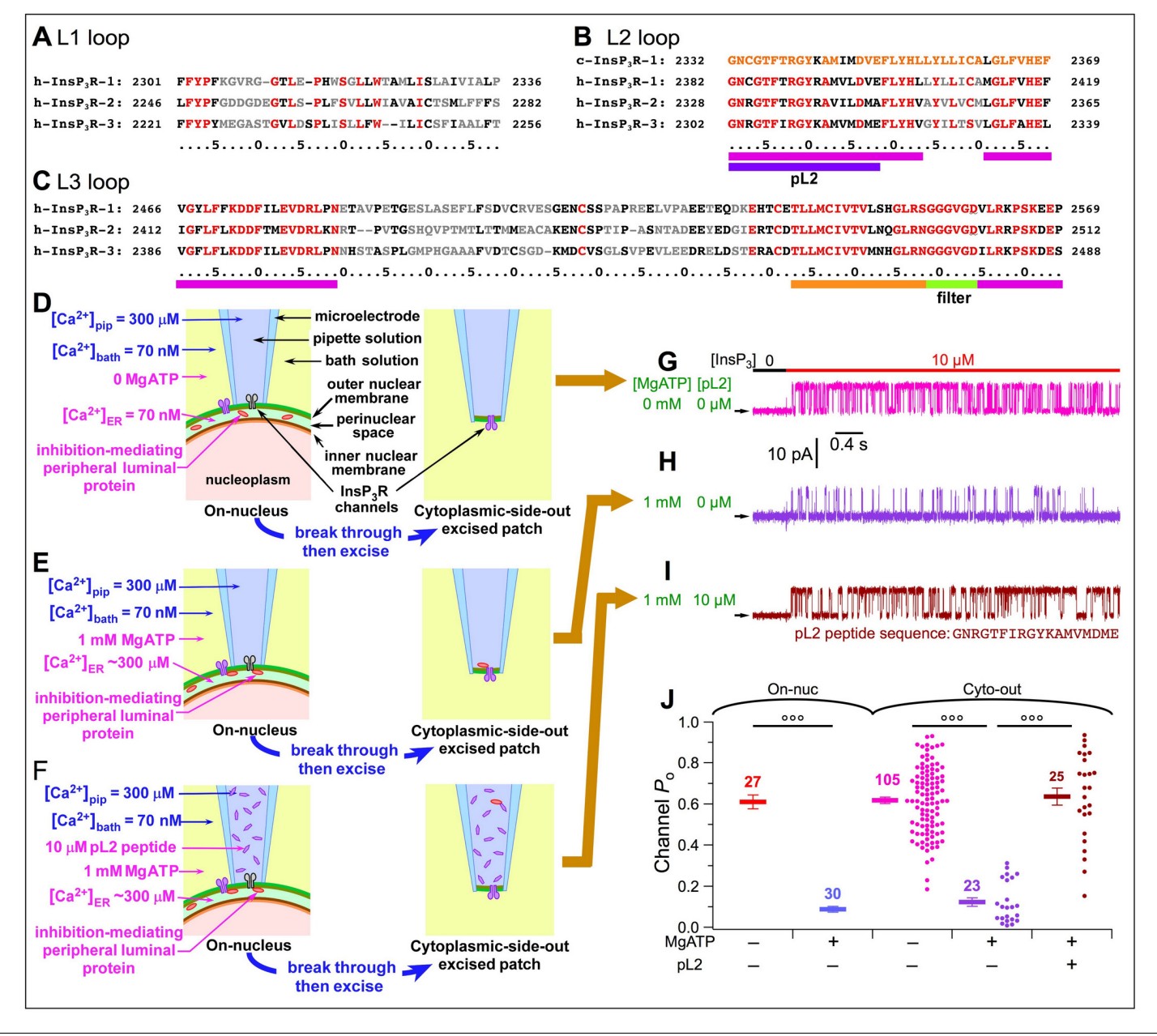

**Figure 5.** Identification of the InsP₃R region involved in $[Ca^{2+}]_{ER}$ regulation of channel activity. (A–C) Sequences of L1 (A), L2 (B) and L3 (C) loops of human InsP₃R isoforms exposed to ER lumen according to h-InsP₃R-1 cryo-EM structure in *Fan et al., 2015*. Residues conserved over all three InsP₃R isoforms are in red; similar residues in black; and different residues in grey. Highly-conserved sequences: magenta stripes. Sequence of pL2 (partial L2) peptide used in experiments is from h-InsP₃R-3: purple stripe. L2 loop sequence of chicken InsP₃R-1 also shown (B, top line) revealing highly-conserved L2 loop sequence. (D–F) Cartoons showing $[Ca^{2+}]_{ER}$-dependent interaction between luminal part of InsP₃R and a peripheral protein in ER lumen in various pipette and bath solutions, during on-nuc or cyto-out nuclear patch-clamp experiments, as labeled. (G–I) Typical InsP₃R-3 current traces in cyto-out experiments with 0- or 1 mM MgATP in bath, and 0- or 10 μM pL2 peptide in pipette solution, as tabulated. Traces in (G), (H) and (I) recorded in experiments depicted in (D), (E) and (F), respectively. In this and all subsequent cyto-out experiments, $V_{app}$ = +30 mV and channels were activated by perfusion solutions containing optimal 2 μM $Ca^{2+}_{i}$, saturating 10 μM InsP₃ and 0.5 mM $ATP^{4-}$. (J) $P_o$ of individual current traces (circles) and averages and s.e.m. (horizontal bars) observed under conditions as tabulated. For comparison, averages and s.e.m. of $P_o$ in on-nuc experiments (red and blue horizontal bars) from *Figure 3J* are shown.

marker calnexin in the nuclear envelope and in cytoplasmic puncta (*Figure 6A*). ANXA1 localization to the nuclear envelope was also observed in HeLa and Panc-1 cells (*Figure 6B–D*). The inability to observe ANXA1 throughout the ER could be the result of its low expression in the bulk ER lumen. Therefore, to further establish localization to the ER lumen, we performed capture ELISA of secreted proteins. The non-secreted β-actin was detected only in the lysate (*Figure 6E*, blue), indicating that the HEKtsA201 cells remained healthy and intact during the 3 day incubation period. In contrast, the signal for extracellular ANXA1 was robust (*Figure 6E*, red), suggesting that ANXA1 was present in the ER lumen en route to be secreted.

## Luminal ANXA1 can mediate $[Ca^{2+}]_{ER}$-dependent InsP$_3$R channel inhibition

We examined direct effects of ANXA1 on channel gating in cyto-out patch-clamp experiments with pipette solutions (bathing the luminal side) containing 1 µM recombinant full-length human ANXA1 under conditions that prevented permeant $Ca^{2+}$ feed-through effects (*Figure 7—figure supplement 1*). ANXA1 was without effect on maximally-stimulated channel $P_o$ ($\approx$ 0.65) in $[Ca^{2+}]_{ER}$ between 70 nM and 55 µM (*Figure 7A–C and F*). In contrast, with $[Ca^{2+}]_{ER}$ >55 µM, channel $P_o$ dropped abruptly with increasing $[Ca^{2+}]_{ER}$ (*Figure 7D–F*). Fitting the $P_o$ data with a simple inhibitory Hill equation (*Figure 7F*):

$$P_o = P_{max}\left\{1 + \left(\frac{[Ca^{2+}]_{ER}}{K_{inh}}\right)^{H_{inh}}\right\}^{-1} \tag{1}$$

showed that the half-maximal inhibitory $[Ca^{2+}]_{ER}$ ($K_{inh}$) is 260 ± 14 µM with Hill coefficient ($H_{inh}$) of 2.3 ± 0.3 (*Figure 7F*). The observed $K_{inh}$ is within observed physiological levels of $[Ca^{2+}]_{ER}$ (~100 µM

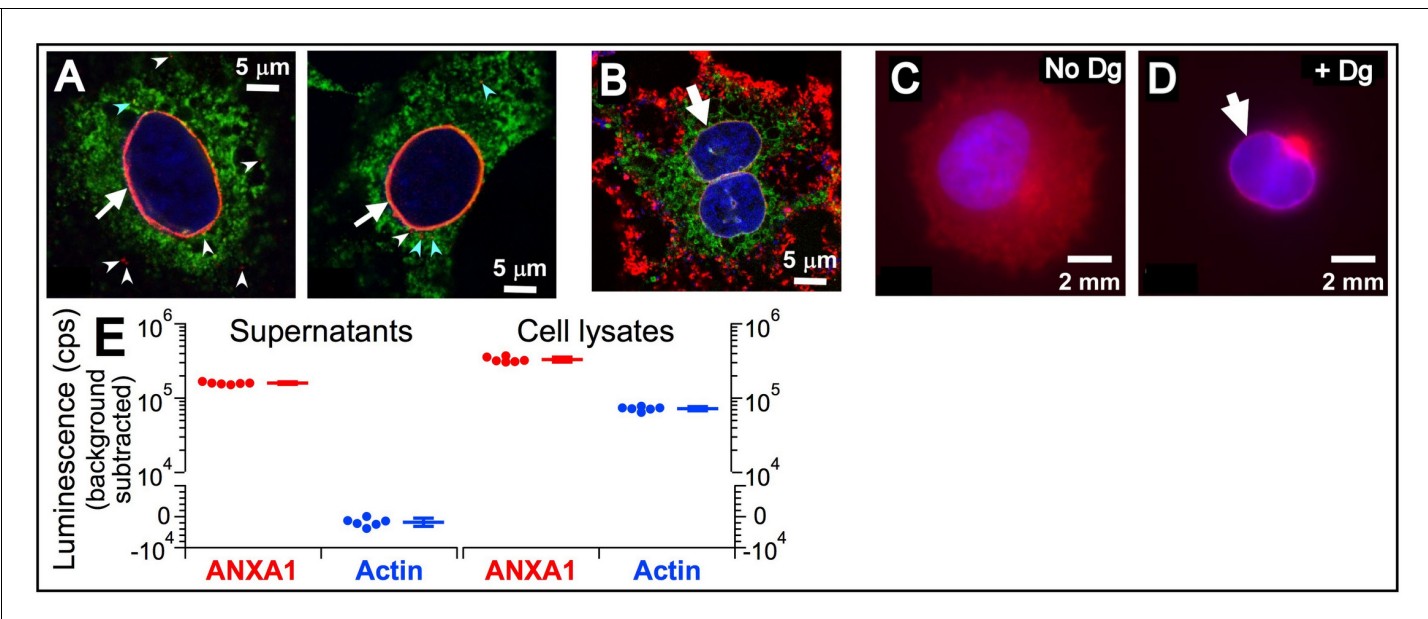

**Figure 6.** Localization of AnxA1 in the ER lumen. (A–B) Confocal images of permeablized A549 (A) and HeLa (B) cells. Native ANXA1 (red), calnexin (green), nuclei (blue). ANXA1 co-localized with calnexin in nuclear envelope (white arrow). ANXA1 also found in puncta without (white arrowhead) or with (cyan arrowhead) calnexin. (C–D) Confocal images of intact (C) and digitonin permeabilized (D) Panc-1 cells. Native ANXA1 (red), nuclei (blue). White arrow in (D) indicates nuclear envelope labeled with ANXA1. (E) ANXA1-(red) and β-actin-(blue) capture ELISA of supernatant (left) and whole-cell lysate (right). Dots: signals detected in individual assays (*n* = 6); horizontal bars: mean, s.e.m.

The online version of this article includes the following source data and figure supplement(s) for figure 6:

**Source data 1.** Abundance of annexin proteins (quantified as total spectral counts by mass spectrometry analysis) detected in experiment eluates collected from magnetic beads covalently linked to peptides with modified pL2 sequence, and in control eluates collected from beads linked to peptide with scrambled modified pL2 sequence.

**Figure supplement 1.** Pull-down of peripheral luminal proteins that interact with IP$_3$R-3 L2 loop.

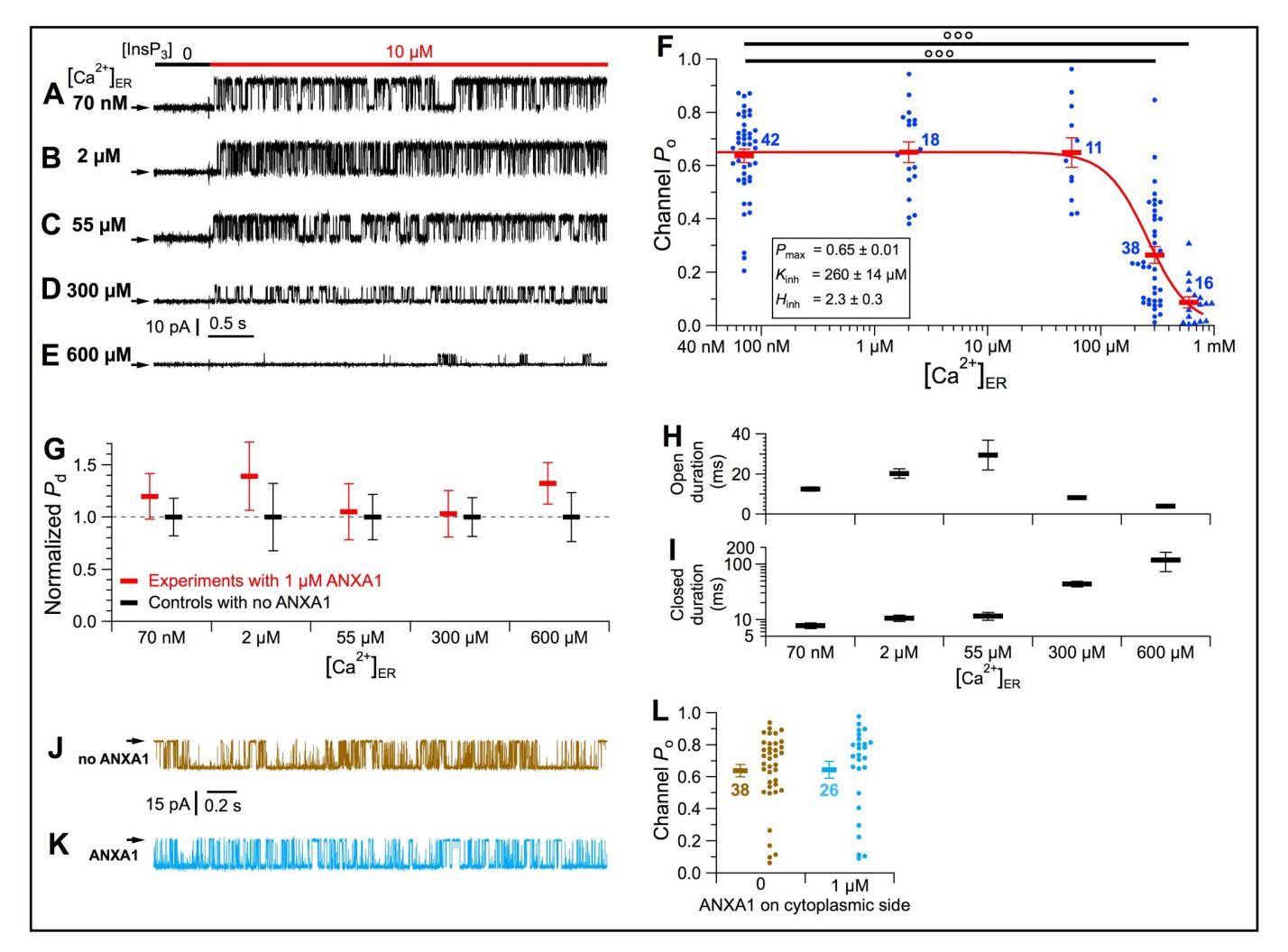

**Figure 7.** $[Ca^{2+}]_{ER}$ inhibition of InsP$_3$R-3 channel activity is mediated by a specific interaction between the channel and ANXA1 in the ER lumen. (A–E) Typical current traces of InsP$_3$R-3 channels in cyto-out configuration with 1 µM ANXA1 and various $[Ca^{2+}]_{ER}$ in pipette solution. In C–E with $[Ca^{2+}]_{ER}$ >40 µM, channel conductance reduced due to permeant-ion block. (F) $P_o$ of individual current traces (blue symbols) and averages and s.e.m. (red horizontal bars) observed under conditions in (A–E). Red curve: least-squares fit to average $P_o$ at various $[Ca^{2+}]_{ER}$ using simple inhibitory Hill equation with parameters tabulated. (G) Averages and s.e.m. of normalized probability of detection of InsP$_3$R channels ($P_d$) in cyto-out experiments with 0 (black) or 1 µM (red) ANXA1 in pipette solutions with various $[Ca^{2+}]_{ER}$. (H–I) Averages and s.e.m. of InsP$_3$R-3 channel open (H) and closed (I) durations in cyto-out patch-clamp experiments in (F). (J–K) Typical current traces in on-nuc patches with pipette (cytoplasmic) solutions containing 2 µM $Ca^{2+}{}_i$ and 10 µM InsP$_3$, with 0 (J) or 1 µM (K) ANXA1. Bath solution: 70 nM $Ca^{2+}{}_{ER}$ with 0-MgATP. (L) Po of individual current traces (brown: no ANXA1; light blue: 1 µM ANXA1 on cytoplasmic side) and averages and s.e.m. (respective horizontal bars) observed under conditions in (J–K).

The online version of this article includes the following figure supplement(s) for figure 7:

**Figure supplement 1.** Abrogation of $Ca^{2+}$ feed-through effect on InsP$_3$R channel activity by buffering $[Ca^{2+}]_i$ with a high concentration (10 mM) of the $Ca^{2+}$ chelator HEDTA in the cytoplasmic solution.

to 1 mM) (***Zampese and Pizzo, 2012***) and $K_d$ of ANX binding to $Ca^{2+}$. Without affecting the number of channels observed (***Figure 7G***), ANXA1 inhibited $P_o$ mainly by increasing channel-closed durations (***Figure 7H–I***).

ANXA1 is primarily localized to the cytoplasm. However, ANXA1 on the cytoplasmic side of the InsP$_3$R was without affect (***Figure 7J–L***). Thus, ANXA1 inhibits InsP$_3$R activity exclusively from the luminal side.

## ANXA1 inhibits InsP₃R channel gating by interacting with the L2 loop region

To determine if ANXA1 inhibition is mediated by interaction with the L2 loop, we performed cyto-out patch-clamp experiments with the pipette solution facing the luminal aspect containing 1 μM ANXA1 and 40 μM of peptides with either the L2 sequence or a scrambled L2 (sL2) sequence (*Figure 8A–B*, respectively) to compete with the InsP₃R for binding to ANXA1 (*Figures 5J* and *7F*). The L2 peptide, but not the sL2 peptide, completely abrogated inhibition of InsP₃R activity by ANXA1 (*Figure 8C*). This suggests that suppression of channel activity by ANXA1 is specifically mediated by the L2 region of the InsP₃R.

## Native PLP-mediated and recombinant ANXA1-mediated [Ca²⁺]$_{ER}$-dependent InsP₃R channel inhibition have similar features

In sub-physiological 70 nM Ca²⁺$_{ER}$, neither the native PLP (*Figure 9*, Lane 2) nor recombinant ANXA1 (*Figure 9*, Lane 5) suppressed channel activities, since $P_o$ observed were indistinguishable from those with no PLP on the luminal side (*Figure 9*, Lane 1). In 300 μM Ca²⁺$_{ER}$, both native PLP (*Figure 9*, Lane 3) and luminal ANXA1 (*Figure 9*, Lane 6) inhibited the $P_o$. Inhibition by PLP or ANXA1 was completely abrogated by L2 loop peptides (*Figure 9*, Lanes 4 and 7, respectively). Such similarities suggest that ANXA1 is the major PLP mediating [Ca²⁺]$_{ER}$-dependent inhibition of InsP₃R channels.

Although native PLP and luminal ANXA1 both inhibit InsP₃R activity, $P_o$ of PLP-inhibited InsP₃R channels (*Figure 9*, Lane 3) was lower than that of ANXA1-inhibited channels (*Figure 9*, Lanes 6 and 8). However, in higher [Ca²⁺]$_{ER}$ (600 μM), ANXA1-inhibited channel $P_o$ (*Figure 9*, Lane 9) was comparable to the native PLP-inhibited $P_o$ (*Figure 9*, Lane 3). This suggests that the luminal concentration of endogenous ANXA1 is effectively higher than 1 μM, and/or that other factor(s) enhance the [Ca²⁺]$_{ER}$ sensitivity of the channel to inhibition by ANXA1.

## Annexin A1 specifically mediates [Ca²⁺]$_{ER}$-dependent InsP₃R channel inhibition

Multiple annexins were identified in our mass spectroscopy analysis, including ANXA6, ANXA2 and ANXA11 (*Figure 6—source data 1*). However, recombinant ANXA6 or ANXA2 had no effect on channel $P_o$ (*Figure 10A–B*). Similar experiments with ANXA11 were precluded by its low solubility. The ANXA6 used had an N-terminal His tag. However, suppression of InsP₃R activities by ANXA1

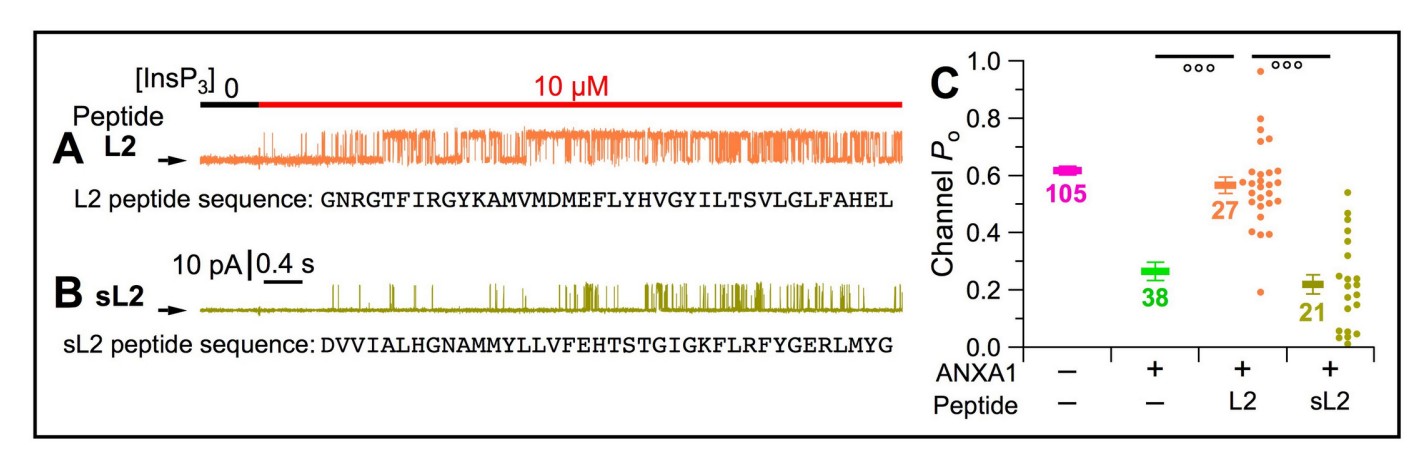

**Figure 8.** L2 peptide, but not scrambled L2 peptide, was able to compete ANXA1 from the InsP₃R channel and restore high channel activity. (A–B) Typical current traces in cyto-out experiments with pipette solutions containing 300 Ca²⁺$_{ER}$, 1 μM ANXA1, and 40 μM of L2 (A) or scrambled L2 (sL2) (B) peptides. Peptide sequences shown below corresponding traces. (C) $P_o$ of individual current traces (orange: L2 peptide; yellow: sL2 peptide) and averages and s.e.m. (horizontal bars) observed under conditions in (A–B). For comparison, averages and s.e.m. of $P_o$ in similar cyto-out experiments without peptide, and without (magenta bars) and with (light green bars) ANXA1, from *Figures 4J* and *6F*, respectively.

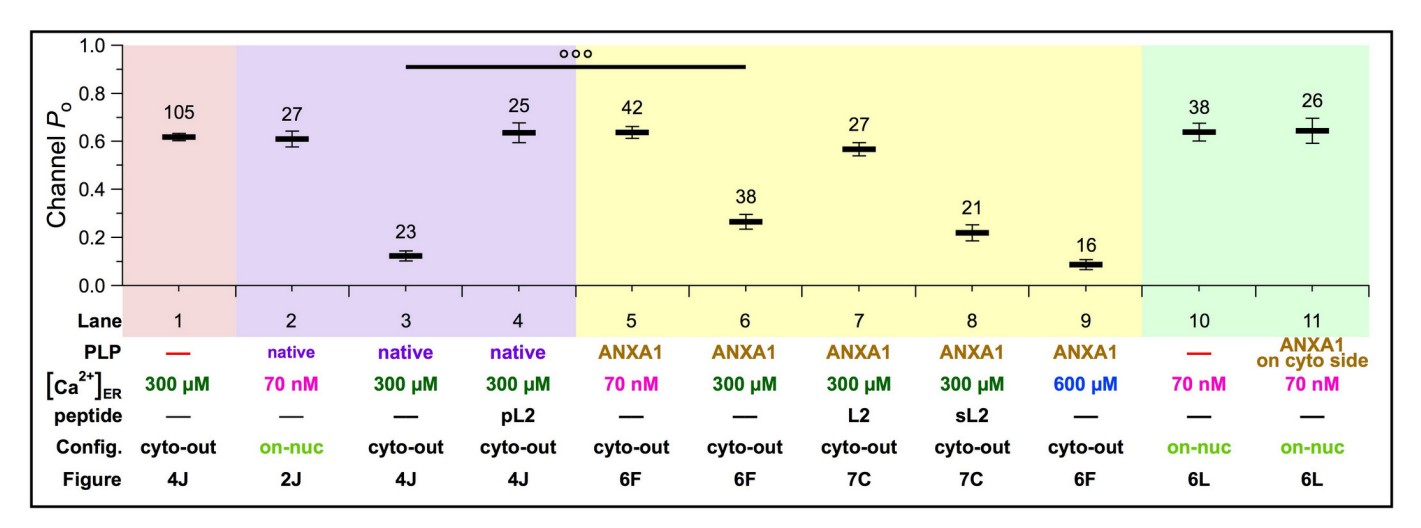

**Figure 9.** Modulation of homotetrameric InsP$_3$R channel activity by native inhibitory peripheral luminal protein or ANXA1 under various experimental conditions: 70 nM, 300 or 600 μM Ca$^{2+}_{ER}$; in the absence or presence of peptides with L2 sequences (pL2, L2 or sL2); and in on-nuc or cyto-out patch-clamp configurations. Averages and s.e.m. of InsP$_3$R-3 $P_o$ are shown as tabulated.

with and without an N–terminal His tag was equivalent (*Figure 10—figure supplement 1*), indicating that the observed effects were not impacted by presence or absence of the tag.

Annexins have a conserved C-terminal 'annexin core' domain with multiple 'annexin-type' Ca$^{2+}$-binding sites (*Gerke and Moss, 2002*). In contrast, the N-terminal regions are diverse (*Raynal and Pollard, 1994*). That ANXA1, but neither ANXA6 nor ANXA2, inhibited gating of InsP$_3$R channels suggests that inhibition is mediated by interaction of its N-terminal domain with the InsP$_3$R. To determine if the N-terminal domain itself can affect channel activity, we included 32 μM N-terminal peptide of human ANXA1 (*Figure 10C*) in the pipette solution with 300 μM Ca$^{2+}_{ER}$, and found that channel $P_o$ was unaffected (*Figure 10D–E*). This suggests that whereas the N-terminal domain may mediate interaction between ANXA1 and the InsP$_3$R L2 region, the ANXA1 C-terminal domain is likely needed to localize ANXA1 to the two-dimensional surface of the ER membrane to increase its local concentration to facilitate N-terminus binding to the InsP$_3$R to suppress channel activity.

## Endogenous ANXA1 inhibits InsP$_3$R-mediated Ca$^{2+}$ release in intact cells

To determine effects of ANXA1 on cytoplasmic Ca$^{2+}$ signaling, we used siRNA to reduce ANXA1 protein expression after 48 hr to ~35% of WT levels in HEKtsA201 cells (*Figure 11A–B*), and measured magnitude ($\Delta R_{max}$) and rate of change ($1/\tau$) of Fura-2 fluorescence ratio in populations of cells responding to the muscarinic receptor agonist carbachol (*Figure 11C*) in a bath solution containing 1.5 mM CaCl$_2$. These parameters capture the initial InsP$_3$R-mediated Ca$^{2+}$-release phase of the response to agonist. Both parameters were enhanced specifically in the *ANXA1* siRNA-treated cells (*Figure 11D–E*, respectively), suggesting that endogenous ANXA1 inhibits InsP$_3$R-mediated Ca$^{2+}$ release.

We also measured histamine-induced changes in Fura-2 fluorescence ratio (R/R$_0$) in individual *ANXA1* and non-targeting (N-T) siRNA-treated HeLa cells (*Figure 11F*). Saturating (100 μM) histamine elicited similar responses in both kinds of siRNA-treated cells (*Figure 11H*). In contrast, in sub-saturating (10 μM) histamine, an increase in both the fraction of responding cells (*Figure 11G*) and the maximal increase in R/R$_0$ was observed in the *ANXA1* siRNA-treated cells (*Figure 11I*). We classified responding cells as 'oscillatory' or 'non-oscillatory' (*Figure 11K*). In 100 μM histamine, the

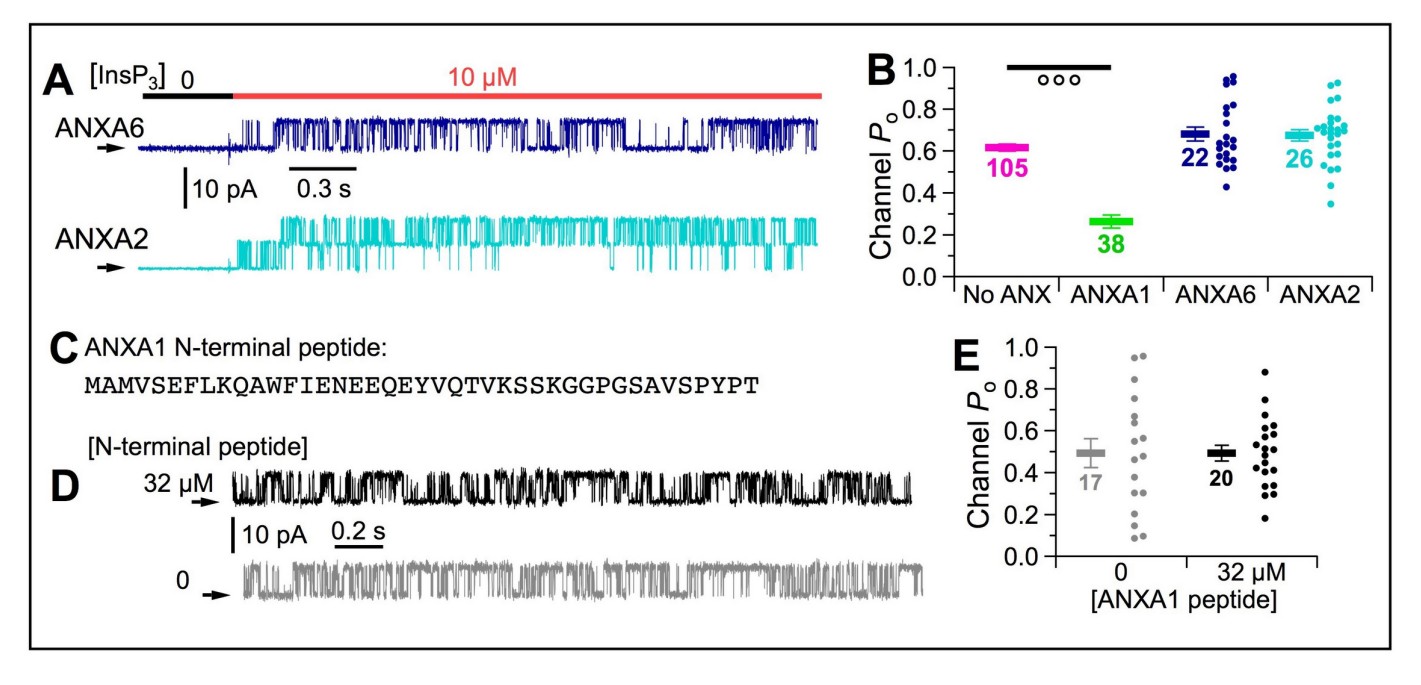

**Figure 10.** $[Ca^{2+}]_{ER}$ inhibition of InsP$_3$R channel activity is specifically mediated by ANXA1 through interaction with the L2 loop of InsP$_3$R. (**A**) Typical current traces of InsP$_3$R-3 in cyto-out experiments with pipette containing 300 µM $Ca^{2+}_{ER}$ and 1 µM ANXA6 or A2. Isolated patches perfused with 2 µM $Ca^{2+}_{free}$, 0.5 mM ATP$^{4-}$ with 0 or 10 µM InsP$_3$ as indicated. (**B**) $P_o$ of individual current traces (circles) and average and s.e.m. (horizontal bars) for cyto-out experiments with ANXA6 (blue) or ANXA2 (cyan). For comparison are averages and s.e.m. of $P_o$ in similar experiments with no ANXA1 (magenta bars) and ANXA1 (green bars), from *Figures 5J* and *6F*, respectively. (**C**) Synthetic peptide with 41 residues of N-terminal domain of ANXA1. (**D**) Typical current traces of InsP$_3$R-3 in cyto-out configuration with pipette containing 300 µM $Ca^{2+}_{ER}$, with or without 32 µM ANXA1 N-terminal peptide. Isolated patches perfused with solution containing 2 µM $Ca^{2+}_i$, 10 µM InsP$_3$ and 0.5 mM ATP$^{4-}$. (**E**) $P_o$ of individual current traces (circles) and average and s.e.m. (horizontal bars) for cyto-out experiments with (black) or without (grey) 32 µM ANXA1 N-terminal peptides.

The online version of this article includes the following figure supplement(s) for figure 10:

**Figure supplement 1.** Presence of a His tag at the N-terminus of the ANXA1 has no effect on its ability to inhibit InsP$_3$R channel activities in 300 µM $[Ca^{2+}]_{ER}$.

fraction of ocillatory cells was indistinguishble between *AnxA1* or N-T siRNA-treated cells; whereas in 10 µM histamine, a significantly higher fraction of *ANXA1* siRNA-treated cells showing either 'random spiking' or 'periodic oscillations' (*Figure 11J*).

We explored this further by visualizing elementary Ca$^{2+}$-release events (Ca$^{2+}$ puffs) generated by clusters of InsP$_3$R stimulated by uncaging of a non-metabolizable InsP$_3$ analogue (*Figure 11L*; *Lock et al., 2018*). More Ca$^{2+}$ puffs were observed in the cells treated with *ANXA1*-siRNA than in cells treated with non-targeting siRNAs (*Figure 11M–N*), with larger amplitudes (*Figure 11O–P*), regardless of whether they were generated by a sub-saturating UV flash (50 ms, *Figure 11M and O*) or a saturating one (150 ms, *Figure 11N and P*). Taken together, these results suggest that endogenous luminal ANXA1 inhibits InsP$_3$R-mediated Ca$^{2+}$ release in vivo, in strong agreement with the strong inhibition by ANXA1 of InsP$_3$R channel activity observed in vitro.

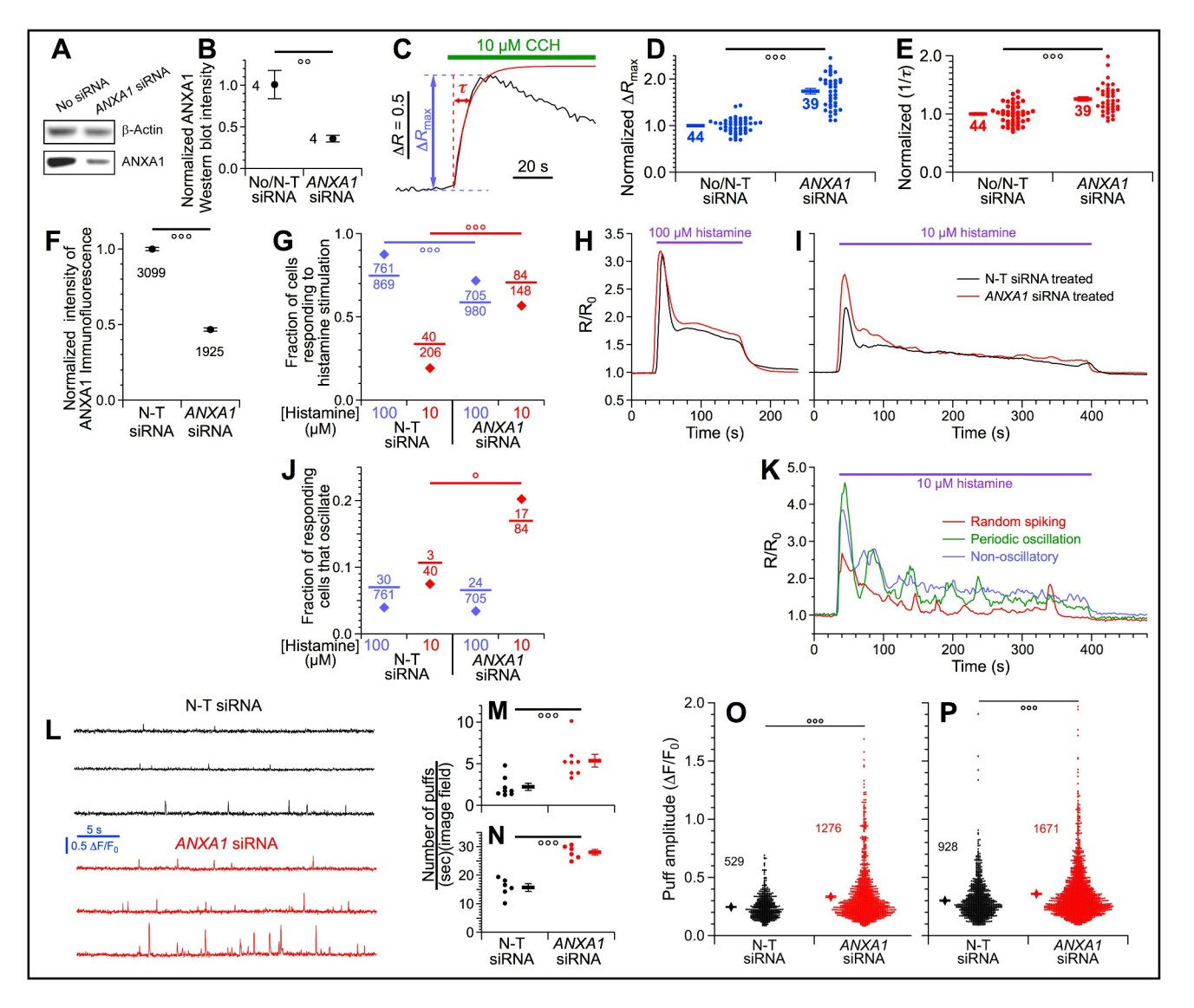

**Figure 11.** Endogenous ANXA1 inhibits InsP$_3$R-mediated Ca$^{2+}$ release. (A) Western blot of ANXA1 and β-actin in lysates from HEKtsA201 cells treated with transfection medium (left) or *ANXA1* siRNA (right) (one of four similar blots shown). (B) Summary of ANXA1 protein knockdown. HEKtsA201 cells treated with *ANXA1* siRNA (four samples), transfection medium only (two samples) or with non-targeting (N-T) siRNA (two samples). (C) Typical trace of fura-2 fluorescence ratio ($\Delta R$) observed in population of intact HEKtsA201 cells stimulated by 10 µM carbachol (CCH). Maximal rise in $\Delta R$ ($\Delta R_{max}$): blue double arrow; time constant ($\tau$) of single exponential fit to rising phase of fluorescence ratio (red curve) is time for $\Delta R$ to rise to $[1-1/e]$ of $\Delta R_{max}$: red double arrow. (D) Normalized $\Delta R_{max}$ of individual $\Delta R$ traces (circles) and averages and s.e.m. (horizontal bars) for *ANXA1* siRNA-treated (right) and control cells (left). Number of traces tabulated next to horizontal bars. (E) Normalized rate of change of $\Delta R$ ($1/\tau$) for $\Delta R$ traces using convention as (D). (F) ANXA1 immunofluorescence intensity of HeLa cells treated with non-targeting (N-T) or *ANXA1* siRNA. Number of cells tabulated below corresponding circles. (G) Fractions of N-T or *ANXA1* siRNA-treated HeLa cells that responded by ER Ca$^{2+}$ release through InsP$_3$R when stimulated by sub-saturating 10 (red) or saturating 100 (blue) µM histamine. (H–I) Traces of mean normalized ER Ca$^{2+}$ release from N-T (red) or *ANXA1* (black) siRNA-treated HeLa cells responding to 100 µM (H) or 10 µM (I) histamine. (J) Fractions of N-T or *ANXA1* siRNA-treated HeLa cells that oscillated in response to 10 (red) or 100 (blue) µM histamine. (K) Selected traces showing different kinds of Ca$^{2+}$ signals in *AnxA1* siRNA-treated HeLa cells responding to sub-saturating 10 µM histamine. (L) Typical fluorescence amplitude ($\Delta F/F_0$) traces showing local Ca$^{2+}$ release events (puffs) in HEK293 cells treated with N-T (black) or *ANXA1* (red) siRNA. Cells stimulated by photolysis of caged i-InsP$_3$ using sub-maximal 50 ms UV flash. (M–P) Puffs generated by 50 ms UV flash were subsequently observed for 30 s in eight imaging fields (M); puffs generated by maximal 150 ms UV flash and subsequently observed for 10 s in six imaging fields (N). Dots indicate numbers of puffs observed for N-T (black) and *ANXA1* (red) siRNA-treated cells. Means and s.e.m. indicated by bars. (O–P) Dot plots of $\Delta F/F_0$ of individual Ca$^{2+}$ puffs observed in N-T (black) and *ANXA1* (red) siRNA-treated cells, generated by 50 ms (O) and 150 ms (P) UV flashes, respectively. Means and s.e.m. indicated by diamonds and bars, respectively.

## Discussion

### A novel mechanism of strong regulation of InsP$_3$R channel activity by ER [Ca$^{2+}$]

Our study has established that physiological [Ca$^{2+}$]$_{ER}$ in the normally-replete ER lumen strongly inhibits the activity of the InsP$_3$R Ca$^{2+}$-release channel, with consequent effects on InsP$_3$-mediated Ca$^{2+}$ signals in cells. This inhibitory effect is not mediated by direct Ca$^{2+}$ binding to luminal sites on the InsP$_3$R nor by flux-through effects of Ca$^{2+}$ on cytoplasmic regulatory sites. Rather, it is mediated by a direct interaction of the InsP$_3$R with a lumen-localized membrane-associated protein, which is likely to be ANXA1.

We first established that high [Ca$^{2+}$]$_{ER}$ strongly inhibits InsP$_3$R channel activity by a mechanism involving a putative ER-luminal peripheral protein. To identify the protein, we focused on a segment that is highly conserved in all three InsP$_3$R isoforms and which, according to a 2015 cryo-EM structure (*Fan et al., 2015*), is located as a luminal loop (L2) between transmembrane helices TM3 and TM4 of InsP$_3$R-1. Interestingly, the L2 sequence was assigned to a different location in a recent cryo-EM structure of InsP$_3$R-3 (*Paknejad and Hite, 2018*). Nevertheless, we discovered ANXA1 by pull-down with the pL2 peptide, and inhibition of channel activity by high [Ca$^{2+}$]$_{ER}$ was relieved by inclusion of L2 peptides in the ER lumen. These experiments establish that the inhibition of InsP$_3$R activity is mediated through a protein interaction with the InsP$_3$R L2 region. The structures in *Paknejad and Hite, 2018* have closed channel pores despite binding of InsP$_3$ and Ca$^{2+}$. Of note, ANXA1 reduces, but does not completely abrogate activity of the InsP$_3$R. It is possible that the topology of the L2 region is different in functionally distinct channel-activation states, with some states enabling it to interact in the ER lumen with ANXA1.

Whereas the predominance of our data suggests that the luminal protein responsible for InsP$_3$R inhibition in high [Ca$^{2+}$]$_{ER}$ is ANXA1, some results give us pause. First, we have been unable to co-immunoprecipitate (co-IP) the two proteins from cell lysates. There are many reasons why co-IP can fail, including low expression of ANXA1 in the ER lumen, particularly in comparison with its strong expression in the cytoplasm; and that the interacting regions in InsP$_3$R and ANXA1 are small, particularly in the case of the InsP$_3$R. Second, whereas we have been able to apply immunofluorescence microscopy to localize ANXA1 to the nuclear envelope in different permeabilized cells with cytoplasmic ANXA1 washed out, localization to the ER has been unconvincing (*Figure 3A–E*). This may reflect a very low abundance of luminal ANXA1 in the peripheral ER that escapes detection. Third, effects of ANX1A knockdown on intracellular Ca$^{2+}$ signaling were modest (*Figures 11D–E,G–J,M–P*). This could be due to the fact that knockdown reduced the ANXA1 expression level by only about 65% (*Figure 11B and F*). Taken together, these results may suggest that ANXA1 is not the *bona fide* PLP. Other ER-luminal proteins have also been reported to regulate InsP$_3$R activity, but their effects are all very different from those of the PLP defined here. Direct interaction with chromogranins enhances activity (*Yoo and Lewis, 2000*), and BiP/Grp78 promotes tetramerization of InsP$_3$R-1, thereby enhancing Ca$^{2+}$ release (*Higo et al., 2010*). In contrast, the PLP here inhibits InsP$_3$R channel activity. Thioredoxin ERp44 interacts directly with InsP$_3$R to suppress channel activity (*Higo et al., 2005*). However, ERp44-mediated inhibition is reduced as [Ca$^{2+}$]$_{ER}$ increases beyond 100 μM, a [Ca$^{2+}$]$_{ER}$ dependence opposite to that of the PLP defined here. Furthermore, ERp44 interacts specifically with InsP$_3$R-1, whereas the PLP here inhibits all InsP$_3$R isoforms.

Annexins are defined by a conserved C-terminal Ca$^{2+}$ binding domain (*Gerke and Moss, 2002*). Although the intrinsic affinity of 'annexin-type' Ca$^{2+}$ binding sites is low ($K_d$ up to mM range) (*Raynal and Pollard, 1994*), the presence of phospholipids, especially those with negatively-charged headgroups, strongly enhances Ca$^{2+}$ affinity (*Gerke and Moss, 2002*). Conversely, the affinity of annexins for acidic phospholipids increases dramatically in the presence of high [Ca$^{2+}$]$_{free}$ (*Raynal and Pollard, 1994*; *Gerke and Moss, 2002*). This feature is likely the mechanism underlying the [Ca$^{2+}$]$_{ER}$ dependence of ANXA1 inhibition of InsP$_3$R activity. In low [Ca$^{2+}$]$_{ER}$, ANXA1, having low affinity for phospholipids, remains in the bulk ER lumen, rendering an interaction between ANXA1 and InsP$_3$R channels that are confined to the ER membrane highly unfavorable entropically. With no ANXA1 interaction, the channel can gate robustly. In contrast, in high [Ca$^{2+}$]$_{ER}$ (>100 μM, *Figure 7F*), the ANXA1 C-terminus binds to phospholipids with high affinity and become restricted to the surface of the ER membrane. This substantially increases the chances for ANXA1 to interact with the InsP$_3$R, which promotes channel inhibition.

## Physiological implications of ANXA1 regulation of the InsP₃R

Inhibition of InsP$_3$R by the PLP is very potent. With $[Ca^{2+}]_{ER}$ sufficiently high ($\geq$300 µM), the PLP and ANXA1 strongly suppressed channel activation even in optimal concentrations of cytoplasmic $Ca^{2+}$ and InsP$_3$, such that channel $P_o$ is limited to ~0.1 (*Figure 7F*), merely 15% of the maximal $P_o$ elicited by saturating $[InsP_3]$ in absence of PLP (ANXA1) (*Vais et al., 2012*). This powerful effect is comparable in magnitude to the maximal inhibition of InsP$_3$R activity by high (50 µM) $[Ca^{2+}]_i$ (*Vais et al., 2012*). Furthermore, the range of $[Ca^{2+}]_{ER}$ over which this regulation occurs is within the physiological range. $[Ca^{2+}]_{ER}$ in fully-replete stores has been measured to be 250 µM – 1 mM (*Zampese and Pizzo, 2012*). We observed strong channel inhibition at 300 µM, with more profound inhibition at 600 µM. During agonist stimulation, bulk $[Ca^{2+}]_{ER}$ falls to concentrations that activate STIM1, which has an apparent luminal-$Ca^{2+}$ affinity of 200–400 µM (*Suzuki et al., 2014*; *Stathopulos et al., 2008*). We observed apparent $Ca^{2+}$ affinity of ANXA1-mediated inhibition of ~250 µM, suggesting that relief of inhibition will occur under physiological conditions that activate STIM1 and store-operated $Ca^{2+}$ entry.

It has been suggested that most InsP$_3$R channels (~95%) in cells are unresponsive to experimentally-induced elevations of $[InsP_3]$, with responsive channels limited to those in immobile clusters from which $Ca^{2+}$ blips and puffs originate (*Lock et al., 2019*; *Keebler and Taylor, 2017*; *Thillaiappan et al., 2017*). The mechanisms that silence the majority of InsP$_3$R are unknown, but our results indicate that the PLP defined here (ANXA1) could play a role. In this scenario, most InsP$_3$R are normally associated with the PLP, rendering them unresponsive to InsP$_3$. However, channels in some clusters may lose their association with the PLP, possibly because of a unique phospholipid composition of the ER membrane there, or because the L2 sequence is not exposed in InsP$_3$R channels in clusters, or for other reasons. Channels in such clusters would be much more responsive to InsP$_3$ and 'licensed' to respond even with the ER fully replete with $Ca^{2+}$. In addition, the strong inhibition of InsP$_3$R channel activity by ANXA1 could possibly help shape intracellular InsP$_3$-induced $Ca^{2+}$ signals. PLP (ANXA1) inhibition will reduce the amount of $Ca^{2+}$ released by shortening the open durations of activated channels (*Figure 7H*) and reducing their frequency by lengthening their closed durations (*Figure 7I*). With fewer $Ca^{2+}$ release events with reduced magnitude, coordination by released $Ca^{2+}$ from channels in a cluster to generate $Ca^{2+}$ puffs can be hampered, preventing puffs from being organized into global $Ca^{2+}$ signals. Such suppression of $Ca^{2+}$ release by the PLP could ensure spatial fidelity of local $Ca^{2+}$ signals and be a fail-safe mechanism to prevent excessive, detrimental release of $Ca^{2+}$ from the ER. At reduced $[Ca^{2+}]_{ER}$, the PLP regulation of InsP$_3$R activity could also be a mechanism to preserve fidelity of $Ca^{2+}$ signals. As $[Ca^{2+}]_{ER}$ drops, the $[Ca^{2+}]$ gradient across the InsP$_3$R is reduced, and the $Ca^{2+}$ flux it drives through the pore decreases. Between 100 and 600 µM $[Ca^{2+}]_{ER}$, the six-fold reduction in $Ca^{2+}$ flux could be compensated by the ~6.6 fold increase in the $P_o$ of the InsP$_3$R channels (from ~0.09 to 0.6) as PLP inhibition of channel activity is alleviated (*Figure 7F*). This could allow the amount of $Ca^{2+}$ released to be maintained despite the drop in $[Ca^{2+}]_{ER}$ to preserve $Ca^{2+}$ signals required to regulate downstream cellular processes.

## Limitations of the current study

1. As noted, co-immunoprecipitation of ANXA1 with the InsP$_3$R was not successful. Further studies, using different methods and antibodies should be performed to determine whether the proteins physically interact.
2. As noted, ANXA1 was secreted into the bath (*Figure 6E*), but its localization in permeabilized cells was evident only in the nuclear envelope (*Figure 6A–D*), and not in the peripheral ER. On the other hand, AnxA1 was initially identified as a potential protein that interacts with the InsP$_3$R through in-vitro pull-down assays and mass spectrometry, using ultra-centrifuged ruptured ER microsome fraction devoid of nuclei. Additional studies employing different antibodies and imaging techniques, including super-resolution light microscopy and electron microscopy should be performed to establish the presence of ANXA1 in the lumen of the entire ER.
3. We demonstrated increased channel $P_o$ in the presence of the pL2 peptide and concluded that it was caused by the peptide reversal of the inhibition by luminal $Ca^{2+}$. A control experiment to demonstrate that the pL2 peptide does not increase channel $P_o$ under conditions with reduced luminal $[Ca^{2+}]$ should be performed. We note however that the channel $P_o$ is already

very high under conditions of reduced luminal [$Ca^{2+}$]. In a control experiment, we could change the conditions, for example use lower [$InsP_3$] to observe channels with a low channel $P_o$ in conditions of reduced luminal [$Ca^{2+}$].

# Materials and methods

## Key resources table

| Reagent type (species) or resource | Designation | Source or reference | Identifiers | Additional information |
|---|---|---|---|---|
| Peptide, recombinant protein | Anx A1 | Abcam | ab86446 | |
| Peptide, recombinant protein | Anx A1 with N-terminal His tag | Abcam | ab184588 | |
| Peptide, recombinant protein | Anx A2 | Abcam | ab93005 | |
| Peptide, recombinant protein | Anx A6 | Abcam | ab92934 | |
| Peptide, recombinant protein | L2 peptide (GNRGTFIRGYKAMVMDME FLYHVGYILTSVLGLFAHEL) | Peptide 2.0 | custom | |
| Peptide, recombinant protein | sL2 peptide (DVVIALHGNAMMYLLVFEH TSTGIGKFLRFYGERLMYG) | Peptide 2.0 | custom | |
| Peptide, recombinant protein | pL2 peptide (GNRGTFIRGYKAMVMDME) | Peptide 2.0 | custom | |
| peptide, recombinant protein | mpL2 peptide (K-Ahx-GNRGTFIRGYRAMVMDME, Ahx stands for 6-aminohexanoate residue) | Peptide 2.0 | custom | |
| Peptide, recombinant protein | smpL2 peptide (K-Ahx-RDYRGMRMIMGETFNVGA, Ahx stands for 6-aminohexanoate residue) | Peptide 2.0 | custom | |
| Peptide, recombinant protein | ANXA1 N-terminal peptide (MAMVSEFLKQAWFIENEEQEYV QTVKSSKGGPGSAVSPYPT) | Peptide 2.0 | custom | |
| Chemical compound, drug | siRNA buffer | Dharmacon | B-002000-UB-100 | |
| Chemical compound, drug | siRNA transfection reagent | Dharmacon | T-2001–02 | |
| Chemical compound, drug | diBrBAPTA (5,5'-dibromo1,2 -bis(o-amino phenoxy)ehane -N,N,N',N'-tetraacetic acid) | Invitrogen | D-1211 | |
| Chemical compound, drug | diBrBAPTA (5,5'-dibromo1, 2-bis(o-amino phenoxy) ehane -N,N,N',N'-tetraacetic acid) | Santa Cruz Biotechnology | sc-2273516 | |
| Chemical compound, drug | 2Hydroxyethyl)ethylenedia minetriacetic acid (HEDTA) | Sigma | H7154 | |

*Continued on next page*

*Continued*

| Reagent type (species) or resource | Designation | Source or reference | Identifiers | Additional information |
|---|---|---|---|---|
| Chemical compound, drug | inositol 1,4,5-trisphosphate | Invitrogen | I-3716 | |
| Chemical compound, drug | inositol 1,4,5-trisphosphate | Santa Cruz Biotechnology | sc-201521 | |
| Chemical compound, drug | NHS-activated magnetic beads | Pierce | 88826 | |
| Chemical compound, drug | Protein A Dynabeads | ThermoFisher | 10006D | |
| Chemical compound, drug | Anti-Flag M2 agarose beads | Sigma | A2220 | |
| Chemical compound, drug | Fura-2 AM | Molecular Probes | I-1225 | |
| Chemical compound, drug | siGLO Red transfection indicator | Dharmacon | D-001630-02-05 | |
| Chemical compound, drug | Cal-520/AM | AAT Bioquest | 21130 | |
| Chemical compound, drug | Caged Ins(1,4,5)P3/PM (caged InsP3) | Sirius Fine Chemical SiChem GmbH | cag-iso-2–145- | |
| Chemical compound, drug | EGTA/AM | ThemoFisher | E1219 | |
| Cell line (Gallus gallus) | DT40 cells (wild-type) | Riken Bioresource Center | RCB1464; RRID:CVCL_0249 | |
| Cell line (Gallus gallus) | DT40-KO cells (with all three InsP$_3$R genes disrupted) | Riken Bioresource Center | RCB1467; RRID:CVCL_4634 | |
| Cell line (Gallus gallus) | DT40-r3 cells | ref. (*Mak et al., 2013b*) in this study | NA | |
| Cell line (Homo-sapiens) | HEK293 cells | ATCC | CRL-1573; RRID:CVCL_0045 | |
| Cell line (Homo-sapiens) | HEK-3KO cells | Kerafast | EUR030; RRID:CVCL_HB82 | |
| Cell line (Homo-sapiens) | HEK293-3KO-r InsP3R-3 cells | this study | NA | |
| Cell line Mus musculus | N2a cells | ATCC | CCL-131; RRID:CVCL_0470 | |
| Cell line (Rattus rattus) | PC12 cells | ATCC | CRL-1721; RRID:CVCL_0481 | |
| Cell line (Homo-sapiens) | tsA201 cells | Sigma-Aldrich | 96121229; RRID:CVCL_2737 | |
| Cell line (Homo-sapiens) | A549 cells | ATCC | CCL-185; RRID:CVCL_0023 | |
| Genetic reagent (*Homo sapiens*) | Anx A1 siRNA | Dharmacon | M-011161-01-0005 | |
| Genetic reagent (*Homo sapiens*) | Non-targeting siRNA | Dharmacon | D-001206-13-05 | |

*Continued on next page*

*Continued*

| Reagent type (species) or resource | Designation | Source or reference | Identifiers | Additional information |
|---|---|---|---|---|
| Antibody | rabbit polyclonal anti -AnxA1 antibody | Proteintech | 21990–1-AP; RRID:AB_11182596 | WB: 1:1000-1:4000 IP: 1:1000-1:10000 IHC: 1:50-1:500 IF: 1:20-1:200 |
| Antibody | mouse monoclonal anti-AnxA1 antibody | ECM Biosciences | AM0211 | ELISA 1:1000 ICC 1:100 IP 1:100 WB 1:1000 |
| Antibody | rabbit polyclonal anti-FLAG antibody | Cell Signaling | 14793S; RRID:AB_2572291 | WB: 1:1000 IP: 1:50 IHC: 1:800 IF: 1:800 FC: 1:1600 Chromatin IP: 1:50 |
| Antibody | goat anti-rabbit IgG (H+L) Cross-Adsorbed Secondary Antibody, Alexa Fluor 568 | Invitrogen | A-11011; RRID:AB_143157 | FC: 1–10 µg/mL ICC: 2 µg/mL IF: 2 µg/mL |
| Antibody | mouse monoclonal anti-calnexin antibody | Chemicon | MAB3126 RRID:AB_143157 | ICC: 1:100-1:250 WB: 1:200-1:2000 IP: 1:200-1:1000 |
| Antibody | rabbit polyclonal anti-β actin antibody | Cell Signaling | 7881S; RRID:AB_1549731 | capture Elisa: 1:100 |
| Antibody | goat polyclonal anti-mouse IgG-HRP antibody | Cell Signaling | 7074S; RRID:AB_2099233 | capture Elisa: 1:1000-1:3000 |
| Antibody | horse polyclonal anti-mouse IgG-HRP antibldy | Cell Signaling | 7076S; RRID:AB_330924 | capture Elisa: 1:1000-1:3000 |
| Antibody | mouse monoclonal anti-βactin antibody | Cell Signaling | 8H10D10; RRID:AB_2242334 | WB: 1:1000 IHC: 1:8000-1:32000 IF: 1:2500-1:10000 FC: 1:200-1:800 |
| Antibody | mouse monoclonal anti-type 3 InsP3R antibody | BD Transduction Laboratories | 610312; RRID:AB_397704 | WB: 1:2000-1:4000 |
| Software, algorithm | QuB | refs. (*Qin et al., 2000*) and (*Bruno et al., 2013*) in this study | | Quantitative single-channel analysis |
| Software, algorithm | IGOR Pro | Wavemetrics | | Figure production and data fitting |
| Software, algorithm | Metamorph v7.7 | Universal Imaging/ Molecular Devices | | Image analysis |
| Software, algorithm | Flika | *Ellefsen et al., 2014* | | Image processing |
| Software, algorithm | Microcal Origin v6.0 | OriginLab | | Data analysis and graphing |
| Software, algorithm | Max Chelator | online freeware | | Calculation of ion concentrations |
| Software, algorithm | MaxQuant, version 1.6.1.0 | online freeware | | Database search |

## Proteins

Recombinant full-length human annexin proteins from Abcam were used in our experiments: ANXA1 (cat. # ab86446 with no tag; ab184588 with N-terminal His tag), Anx A2 (cat. # ab93005 with N-terminal His tag), and Anx A6 (cat. # ab92934 with N-terminal His tag).

## Synthetic peptides

Peptides used in our study: L2 (GNRGTFIRGYKAMVMDMEFLYHVGYILTSVLGLFAHEL), sL2 (DVVIA LHGNAMMYLLVFEHTSTGIGKFLRFYGERLMYG), pL2 (GNRGTFIRGYKAMVMDME), mpL2 (K-Ahx-G NRGTFIRGYRAMVMDME, Ahx stands for 6-aminohexanoate residue), smpL2 (K-Ahx-RDYRGMRMI

MGETFNVGA), and ANXA1 N-terminal peptide (MAMVSEFLKQAWFIENEEQEYVQTVKSSKGGPGSA VSPYPT) were custom synthesized by Peptide 2.0 (Chantilly, VA).

## Generation and maintenance of cell lines

All cell lines used were obtained from commercial sources and were routinely tested and confirmed to be mycoplasma-free via PCR. Wild-type DT40 chicken B cells and DT40-KO cells in which all three endogenous InsP$_3$R genes have been stably ablated were obtained from Riken Bioresource Center, Japan (cell bank # RCB1464 and RCB1467, respectively). Generation of DT40–KO-r–InsP$_3$R–3 cells (DT40-r3 cells) that stably express only recombinant rat type 3 InsP$_3$R in DT40-KO cells was described in *Mak et al., 2005*. Wild type DT40 and DT40-r3 cells used in this study were grown in suspension culture in RPMI 1640 medium with 2 mM L-glutamine (Gibco cat. # 11875–085), supplemented with 10% (v/v) FBS, 1% chicken serum, and 1% Gibco antibiotic-antimycotic, and 1% G418 for selection; at 37°C in 5% CO$_2$. When cell density exceeded $2.5 \times 10^6$/ml, they were sub-cultured to $0.1$–$0.2 \times 10^6$/ml (*Mak et al., 2005*).

Wild type HEK293 cells from ATCC (CRL-1573) and InsP$_3$R-Null HEK-293 cells with all three endogenous InsP$_3$R genes knocked out by CRISPR/Cas9 technology (HEK-3KO) from Kerafest (cat. # EUR030) were cultured as adherent cells in DMEM (with L-glutamine, glucose and sodium pyruvate; Corning cat. # MT10-013-CM), supplemented with 10% FBS and 1% Gibco antibiotic-antimycotic; at 37°C in 5% CO$_2$. Medium was renewed every two to three days. Cells were sub-cultured following instructions provided in ATCC CRL-1573 product sheet.

HEK-3KO cells stably expressing rat InsP$_3$R-3 (HEK–3KO-r–InsP$_3$R–3) were generated from HEK293-3KO cells by transfecting them using Transit LT-1 (Mirus cat. # MIR 2304) following manufacturer's protocols. Using G418 selection, stable clones of HEK293-3KO-r-InsP$_3$R-3 cells expressing various level of recombinant r-InsP$_3$R-3 were generated by limiting dilution.

Mouse N2a cells from ATCC (CCL-131) were cultured as adherent cells in 1:1 mixture of DMEM (high glucose with L-glutamine; Gibco cat. # 11965–092) and Opti-MEM I Reduced Serum Medium (Gibco cat. # 31985–070), supplemented with 5% FBS, 1% penicillin-streptomycin; at 37°C in 5% CO$_2$. Cells were sub-cultured following instructions provided in ATCC CCL-131 product sheet.

Rat PC12 cells from ATCC (CRL-1721) were cultured as adherent cells in F-12K Kaighn's Mod. Nutrient mixture (Corning Catalog # 10–025-CV), supplemented with 15% horse serum, 2.5% FBS, 1% Gibco antibiotic-antimycotic; at 37°C in 5% CO$_2$. Cells were sub-cultured following instructions provided in ATCC CRL-1721 product sheet.

Human HEKtsA201 cells derived from HEK293 cells were obtained from Sigma-Aldrich (cat. # 96121229) and cultured as adherent cells in the same medium as for HEK293 cells, at 37°C in 5% CO$_2$. 70–80% sub-confluent cells were dislodged using 0.25% trypsin solution and split 1:4 to 1:8 as described in ECACC data sheet for HEKtsA201 cells.

A549 human carcinoma cells from ATCC (CCL-185) were cultured using the same protocol as that for HEKtsA201 and HEK293 cells.

Panc-1 human pancreatic epithelioid carcinoma cells from ATCC (CRL-1469) were cultivated as adherent cells in DMEM with L-glutamine, glucose and sodium pyruvate (Corning cat. # MT10-013-CM), supplemented with 10% FBS and 1% Gibco antibiotic-antimycotic; at 37°C in 5% CO$_2$. Cells were sub-cultured following instructions provided in ATCC CCL-1469 product sheet.

## Nuclear patch-clamp electrophysiology

Isolation of intact nuclei from cells, performing nuclear patch-clamp experiments in on-nucleus, cytoplasmic-side-out and luminal-side-out configurations with rapid exchange of ligand conditions are described in detail in *Mak et al., 2013b*; *Mak et al., 2013c*; *Vais et al., 2010b*. All solutions used for nuclear patch-clamp experiments contained 140 mM KCl, 10 mM HEPES (pH 7.3). Solutions on the cytoplasmic side of the nuclear membrane contained 2 μM free [Ca$^{2+}$] ([Ca$^{2+}$]$_{free}$). [Ca$^{2+}$]$_{free}$ was buffered either by 0.5 mM 5,5′-dibromo 1,2-bis(o-aminophenoxy)ethane-N,N,N′,N′-tetraacetic acid (diBrBAPTA, Invitrogen D-1211 or Santa Cruz Biotechnology sc-2273516) in experiments in which [Ca$^{2+}$]$_{free}$ on the luminal side of the membrane ≤300 μM; or by 5 mM diBrBAPTA or 10 mM N-(2-Hydroxyethyl)ethylenediaminetriacetic acid (HEDTA, Sigma H7154) in experiments with luminal [Ca$^{2+}$]$_{free}$ ≥300 μM to eliminate possible permeant Ca$^{2+}$ feed-through effects on InsP$_3$R channel activities (*Vais et al., 2012*). Cytoplasmic solutions also contained 0.5 mM Na$_2$ATP (*Mak et al.,*

*2001*) and either sub-saturating 3 µM or saturating 10 µM [InsP$_3$] (*Vais et al., 2012*) (Invitrogen I-3716 or Santa Cruz Biotechnology sc-201521) to stimulate InsP$_3$R channel gating. The luminal solution with 70 nM [Ca$^{2+}$]$_{free}$ was buffered by 0.5 mM BAPTA; that with 2 µM [Ca$^{2+}$]$_{free}$ was buffered by 0.5 mM diBrBAPTA; that with 55 µM [Ca$^{2+}$]$_{free}$ was buffered by 0.5 mM nitrilotriacetic acid (NTA, Sigma 72559) (*Dweck et al., 2005*). [Ca$^{2+}$]$_{free}$ in these solutions were verified using [Ca$^{2+}$]-sensitive dye fluorimetry. The luminal solution with 300 µM [Ca$^{2+}$]$_{free}$ was buffered by 1.5 mM Na$_2$ATP according to Max Chelator freeware; and that with 600 µM [Ca$^{2+}$]$_{free}$ was made using the activity coefficient of CaCl$_2$ as described in *Vais et al., 2010a*. In the experiments involving MgATP to energize the SERCA, we used a solution containing 140 mM KCl, 10 mM K-HEPES, 0.5 mM K-BAPTA, 60 µM CaCl$_2$, 1 mM MgATP, pH 7.3. Using the online Webmaxc Extended software, we calculated that our solution at 25°C has 0.30 mM free Mg$^{2+}$ (with 73 nM Ca$^{2+}_{free}$, 0.30 mM ATP$^{4-}$, and 0.42 mM free BAPTA).

The nuclear patch-clamp experiments in various patch-clamp configurations (on-nuc, cyto-out and lum-out) using DT40-r3 cells were acquired over an extensive period. To ensure the behaviors of these cells were consistent, we regularly used cells, cultured in identical conditions and at most one day apart from data-generating cells, to perform control experiments in which nuclei were patched in on-nuc or cyto-out configurations under optimal stimulatory conditions (with 10 µM InsP$_3$, 2 µM [Ca$^{2+}$]$_i$ and 0.5 mM ATP$^{4-}$ on the cytoplasmic side of the outer nuclear membrane). Our experience suggested that most newly-thawed DT40-r3 cells were consistent after culturing for 5 days. If the mean channel $P_o$ of a batch of control cells cultured for any time beyond day five was found to be significantly lower than that of previous control cells, that batch was terminated immediately and a new batch of cells was thawed out for new experiments.

InsP$_3$R channel-current traces under constant or ramping applied potential ($V_{app}$) were acquired at room temperature (RT) as described (*Mak et al., 2007*), digitized at 5 kHz, anti-aliasing filtered at 1 kHz. All $V_{app}$ were measured relative to the reference bath electrode regardless of the patch-clamp configuration used. InsP$_3$R channel gating characteristics—number of active channels observed ($N_A$) and open probability ($P_o$)—were derived from current traces using semi-automatic QuB (*Qin et al., 2000*) and fully-automatic (*Bruno et al., 2013*) software and manually using IGOR Pro software (Wavemetrics). Only current traces long enough for $N_A$ to be determined with >99% confidence (*Vais et al., 2010b*) were used for statistical analysis.

## Ca$^{2+}$-dependent affinity-enrichment mass spectrometry
### Obtaining samples of enriched peripheral ER-luminal proteins from bovine hepatocytes
Bovine liver was procured from three cows within three hours of slaughter, cut into large pieces of ~200 g each, rinsed with ice-cold PBS and flash frozen in –80°C freezer within one hr of procurement to be used later. The following protocol was used for processing 200 g of bovine liver. Quantities of reagents used were scaled according to amount of extracted proteins required. Bovine liver was thawed in ice-cold water and diced into small pieces of 0.8 cm$^3$ (~0.5 g). 10 g of cut liver pieces and 19 ml of ice-cold DPBS with Mg$^{2+}$ and Ca$^{2+}$ (Corning cat # 21–030 CM) were put into each of twenty Stomacher 80 strainer bags (Seward cat # BA6040/STR) and processed individually in a Stomacher 80 Biomaster (Seward cat # 0080/000/AJ) at highest setting for 75 s to release intact hepatocytes. Cell suspension was strained through the strainer bags and collected (~500 ml). 250 ml of ACK lysing buffer (Quality Biological cat. # 118-156-101) at RT was mixed into the cell suspension to specifically lyse red blood cells. The process was terminated after 5 min by adding 500 ml DMEM (Corning cat. # 10–013-CV) with 10% FBS followed with gentle shaking. Hepatocyte suspension was centrifuged at 9,600 *g* for 18 min and the supernatant was discarded. Hepatocytes were rinsed twice by resuspending cells in 500 ml HBSS, centrifuging at 9,600 *g* for 18 min, and discarding the supernatant.

Hepatocytes were resuspended in 150 ml of homogenizing solution containing 150 mM KCl, 1 mM MgCl$_2$, 1 mM CaCl$_2$, 20 mM Tris HCl (pH 7.5), 400 µM phenylmethane sulfonyl fluoride (PMSF), and seven cOmplete protease inhibitor cocktail tablets (Roche cat. # 11697498001). Debris in cell suspension was removed with cell strainer. The hepatocytes were homogenized with nitrogen cavitation in an ice-cold large cell disruption vessel (Parr Instrument, Moline Il, model # 4635) using pressure of 1000 p.s.i. with constant stirring for 20 min. Cell homogenate was centrifuged at 3,300 *g* for

15 min to pellet unbroken cells and nuclei. The supernatant was centrifuged at 9,600 $g$ for 15 min to pellet cell debris and mitochondria, and at 105,000 $g$ for 15 min to pellet a crude microsomal fraction. The pellet was resuspended into 40 ml of 1.38 M sucrose solution, placed under a step gradient of 1.0 and 0.86 M sucrose and centrifuged at 300,000 $g$ for 75 min. The smooth and rough ER fractions below the interface between 1.38 M and 1.0 M sucrose solution layers were collected and placed on ice.

To remove ribosomes from the rough ER, ice-cold sodium pyrophosphate (NaPP$_i$)-imidazole HCl (NaPP$_i$-I) solution containing 5 mM NaPP$_i$ and 3 mM imidazole-HCl, pH 7.4 was added dropwise to the collected ER fractions with continuous stirring on ice to the final volume of 100 ml. After a further 15 min of stirring, the mixture was centrifuged at 105,000 $g$ for 37 min to re-pellet the microsomes. After discarding the supernatant, the ER microsome pellet was resuspended in 45 ml of sucrose-NaPP$_i$-I solution containing 0.25 M sucrose, 5 mM NaPP$_i$, and 3 mM imidazole HCl, pH 7.4; centrifuged again at 105,000 $g$ for 37 min; and the supernatant was discarded.

The smooth ER microsome pellet was resuspended with vigorous vortexing in 8 ml of low-Ca$^{2+}$ K$^+$ with protease inhibitor (LCaK-PI) solution containing 150 mM KCl, 0.5 mM EGTA, 0.05% Tween-20, 25 mM Tris HCl (pH 7.5), 400 µM PMSF, one cOmplete protease inhibitor cocktail mini-tablet (Roche cat. #11836170001). The microsome suspension was processed with nitrogen cavitation twice using an ice-cold small cell disruption vessel (Parr Instrument, Moline Il, model # 4639) under 2200 p. s.i. with constant stirring for 20 min each cavitation. The homogenate was centrifuged at 105,000 $g$ for 60 min to pellet the broken ER membrane. The supernatant was collected carefully without disturbing the ER membrane pellet so it contained only peripheral proteins released from the ER lumen in the presence of low [Ca$^{2+}$]$_{free}$ (buffered by 0.5 mM EGTA in the LCaK-PI solution). Total protein concentration in the hepatocyte peripheral ER luminal protein (HPERLP) sample was determined using Bradford protein assay (~1.4–3.0 mg/ml). The HPERLP sample was aliquoted and stored in – 80°C for future use.

## Magnetic-bead pull down of peripheral proteins in the ER lumen that specifically interact with the luminal region of the InsP$_3$R

Based on the pL2 sequence in the L2 region of the InsP$_3$R that is highly conserved among the three InsP$_3$R isoforms (*Figure 5B*), peptides with a modified sequence (mpL2: *K*-Ahx-GNRGTFIRGY<u>R</u>-AMVMDME) were synthesized for pull-down experiments. The Lys residue in pL2 was replaced with Arg (underlined) so that the mpL2 peptides could be coupled to magnetic beads through the only Lys (in italics) at the N-terminus. The unnatural 6-aminohexanoate residue (Ahx) was added to reduce possible steric hindrance that may interfere with the interaction between the peptide and ER luminal proteins. For control, peptides with a scrambled modified sequence (smpL2: *K*-Ahx-RDYRGMRMI MGETFNVGA) were also synthesized. Instructions of the manufacturer were followed to covalently couple the mpL2 and smpL2 peptides to NHS-activated magnetic beads (Pierce cat. # 88826) through the stable covalent link formed by reaction between the primary amine group of the N-terminal Lys of the peptides and the *N*-hydroxysuccinimide ester group on the magnetic beads. Peptide-coupled beads were stored (10 mg/ml) in storage buffer (150 mM NaCl, 25 mM Tris, pH 7.4 with 0.05% Tween-20% and 0.05% sodium azide) at 4°C.

A total of six pull-down procedures were performed in the cold with all apparatus and reagents at 4°C: one procedure using mpL2 peptide-coupled magnetic beads and one control procedure using smpL2 peptide-coupled magnetic beads for each of the three HPERLP samples from individual cows. The same protocol described below was used for each of the six procedures. To obtain sufficient protein for mass spectrometry, eight MACS separation columns (MS columns, Miltenyi Biotec cat. # 130-042-201) were used in each procedure. The MS columns were placed on the OctoMACS separator (Miltenyi Biotec cat. # 130-042-109) mounted on the MACS Multistand (Miltenyi Biotec cat. #130-042-303). To prime the columns, 250 µl of high-Ca$^{2+}$ K$^+$ (HCaK) solution containing 150 mM KCl, 1% NP–40 detergent (Thermo scientific cat. # 85124), 0.8 mM CaCl$_2$, 25 mM Tris HCl (pH 7.5) was added to each. Effluent from the columns was discarded. 320 µl of peptide-coupled magnetic beads were resuspended in storage buffer, rinsed twice with HCaK solution, and resuspended with 290 µl of HCaK solution. 35 µl of the peptide-coupled magnetic bead suspension was added to each of the MS columns. Effluent was examined for presence of magnetic beads. If beads were detected, effluent was returned to the column till no beads were detected. Each column with

magnetic beads were washed with 80 µl of HCaK solution and effluent was discarded. 360 µl of the HPERLP sample was thawed and HCaK solution and 1 M CaCl$_2$ standard solution were added to adjust the total protein concentration in the final HCaK-HPERLP mixture to 1.25 mg/ml, and [CaCl$_2$] to 800 µM ([Ca$^{2+}$]$_{free}$ = 432 µM). 40 µl of the HCaK-HPERLP mixture was added to each column, followed by 80 µl of HCaK-PI solution—HCaK solution with additional 400 µM PMSF and 1% Halt protease inhibitor cocktail (Thermo Fisher Cat. # 78430). The magnetic beads in each column were rinsed once with 2 ml of HCaK-PI solution, and thrice with 2 ml of HCaK solution. Effluent was discarded. 200 µl of low-Ca$^{2+}$ eluding (LCaE) solution with 150 mM KCl, 1% NP–40 detergent, 0.5 mM EGTA, 25 mM Tris HCl (pH 7.5) was added to each column. The eluate from all the columns was collected and frozen at –80°C.

## Loading samples into mass spectrometer

Low-Ca$^{2+}$ eluate proteins from above were precipitated with 5 volumes of acetone held overnight at –20°C, decanted then solubilized in 20 µL of 1 × reducing NuPAGE LDS (Thermo Fisher Scientific) and run into a precast 10% Bis-Tris SDS-PAGE (Thermo Fisher Scientific) for 1.6 cm. After fixing overnight (25% ethanol, 7% acetic acid) the gel was stained with colloidal Coomassie blue dye. Each sample was excised from the gel in 4 (2 × 9 mm) slices and each cut into (1 mm)$^3$ cubes. The gel pieces were destained in a solution containing 50% methanol and 1.25% acetic acid, reduced with 5 mM DTT (Thermo Fisher Scientific), and alkylated with 40 mM IAA (Sigma-Aldrich). The gel pieces were washed with 20 mM ammonium bicarbonate (Sigma-Aldrich) and dehydrated with acetonitrile (Fisher Scientific) twice. Sequencing-grade trypsin (Promega) (5 ng/µL in 20 mM ammonium bicarbonate) was added to the dehydrated-gel pieces and proteolysis was allowed to proceed overnight at 37°C. Peptides were extracted with 0.3% TFA (Fisher Scientific), followed by 50% acetonitrile. The volume of the combined extracts was reduced by vacuum centrifugation and 25% of the extracted peptides were analyzed per LC-MS/MS run (*Glisovic-Aplenc et al., 2017*).

## LC-MS/MS mass spectrometry

Tryptic digests were analyzed by LC-MS/MS on a hybrid LTQ Orbitrap Elite mass spectrometer (Thermofisher Scientific San Jose, CA) coupled with a nanoLC Ultra (Eksigent). Peptides were separated by reverse phase (RP)-HPLC on a nanocapillary column, 75 µm id ×15 cm Reprosil-pur 3 µm, 120A (Dr. Maisch, Germany) in a Nanoflex chip system (Eksigent). Mobile phase A consisted of 1% methanol (Fisher)/0.1% formic acid (Thermo) and mobile phase B of 1% methanol/0.1% formic acid/ 80% acetonitrile. Peptides were eluted into the mass spectrometer at 300 nL/min with each RP-LC run comprising a 90 min gradient from 10% to 25% B in 65 min, 25–40% B in 25 min. The mass spectrometer was set to repetitively scan m/z from 300 to 1800 (R = 240,000 for LTQ-Orbitrap Elite) followed by data-dependent MS/MS scans on the twenty most abundant ions, with a minimum signal of 1500, dynamic exclusion with a repeat count of 1, repeat duration of 30 s, exclusion size of 500 and duration of 60 s, isolation width of 2.0, normalized collision energy of 33, and waveform injection and dynamic exclusion enabled. FTMS full scan AGC target value was 1e6, while MSn AGC was 1e4, respectively. FTMS full scan maximum fill time was 500 ms, while ion trap MSn fill time was 50 ms; microscans were set at one. FT preview mode; charge state screening, and monoisotopic precursor selection were all enabled with rejection of unassigned and 1+ charge states.

## Sequence database search

Raw MS files were processed using MaxQuant, version 1.6.1.0 for identification of proteins. The peptide MS/MS spectra were searched against the UniProtKB/Swiss-Prot Bovine Reference Proteome database, UP000009136 (retrieved on 12 July, 2018), comprising 24,333 entries including isoforms. Fragment ion tolerance was set to 0.5 Da, with full tryptic specificity required and a maximum of two missed tryptic cleavage sites. Precursor ion tolerance was seven ppm. Oxidation of methionine, acetylation of the protein N-terminus and conversion of glutamine to pyroglutamic acid were used as variable modifications. Carbamidomethylation of cysteine was set as a fixed modification. The minimal length required for a peptide was seven amino acids. Target-decoy approach was used to control false discovery rate (FDR). A maximum FDR of 1% at both the peptide and the protein level was allowed. If no unique peptide sequence to a single database entry was identified, the resulting protein identification was reported as an indistinguishable 'protein group'. Protein groups containing

matches to decoy database or contaminant proteins were discarded. The MaxQuant match-between-runs feature was enabled.

## Statistical analysis of mass spectrometry output

We modeled the MS/MS spectral counts for each protein, sample pair using a hierarchical Poisson log-linear model. Our Bayesian method helped circumvent some of the problems that arise with small sample sizes by borrowing strength across proteins to better estimate the parameters in the model. We performed inference to determine the proteins that preferentially bound to the bait peptide found in InsP$_3$R compared to the scrambled, control peptide using the false sign rate paradigm (**Stephens, 2017**).

Let $y_{gct}$ be the number of observed MS/MS spectra for protein $g = 1, ...p$ from cow $c = 1, ..., C$ in sample $t \in \{0, 1\}$, where $t$ is 0 if the bait molecule was the scrambled peptide and one if the bait molecule was the modified InsP$_3$R L2, that is a treated sample. In these data, $C = 3$ and $p=486$. For $\delta_0$ a point mass at 0, we model the data as follows:

$$y_{gct}(\lambda_{gct}) + 1 \sim Poi(\lambda_{gct}) \ (g = 1, ..., p; c = 1, ...C; t = 0, 1)$$
$$\log(\lambda_{gct}) = \log(\mu_c) + \alpha_{gc} + t\beta_g \ (g = 1, ..., p; c = 1, ...C; t = 0, 1)$$
$$\alpha_{gc} \sim N(0. \sigma_c^2) \ (g = 1, ..., p; c = 1, ..., C)$$
$$\beta_g \sim \pi_0 \delta_0 + (1 - \pi_0)N(0, \tau^2)$$
$$\pi_0 \sim Beta(0.5, 0.5)$$
$$\sigma_c^{-2}, . \tau^{-2} \sim Gamma(1, 1) \ (c = 1, ..., C)$$

and set $\mu_c = p^{-1}\Sigma_{g=1}^{p} \frac{y_{gc0}+y_{gc1}+2}{2}$. We added a pseudo count of 1 spectral count to each data point to avoid taking the log of 0. Note that adding a pseudo count only makes it more challenging to identify differences between the treatment and control conditions, and therefore leads to conservative inference. The above model also helped mitigate latent confounding variables and sources of over-dispersion by explicitly accounting for between-cow variation.

The goal was to estimate and perform inference on the treatment effect $\beta_g$ for each protein $g = 1, ..., p$, where we were interested in identifying proteins with $\beta_g > 0$ because these were indicative of proteins that were comparatively enriched in the treated, experimental samples. The local false sign rate for each protein is defined as the posterior probability that protein $g$ did not preferentially bind to the InsP$_3$R peptide:

$$lfsr_g = P(\beta g \leq 0|Y) \ (g = 1, ..., p).$$

The local false sign rates were computed by sampling from the posterior using Markov Chain Monte Carlo. A low LFSR$_G$ value indicates that the posterior probability of incorrectly identifying a protein as being enriched in the experimental samples is small. A threshold of LFSR$_G \leq 0.2$ (corresponding to a global false discovery rate of 5%) was applied to identify proteins with strong preference to bind to the L2 region of the InsP$_3$R.

## Confocal microscopy

Images were collected on a Leica TCS SP8 X system using a 100x/1.4 NA PL APO CS2 objective. Annexin A1 was labeled with rabbit anti-Annexin A1 pAb (Proteintech, 21990–1-AP) and anti-rabbit AlexaFluor568 conjugated pAb (Invitrogen, A11011) was excited using a continuously tunable white light laser set to 568 nm to detect fluorescence emission between 600–700 nm. Calnexin was labeled with mouse anti-Calnexin mAb (Chemicon, MAB3126) and anti-mouse AlexaFluor488 conjugated pAb (Invitrogen, A11001) excited at 488 nm to detect fluorescence emission between 510–550 nm. DAPI was present in the mounting medium used to prepare the slides, excited at 408 nm to detect fluorescence emission between 430–470 nm.

## Capture ELISA to monitor secretion of ANXA1 from HEKtsA201 cells

### Sample preparation

HEKtsA201 cells were seeded at $2 \times 10^6$ cells/10 cm tissue culture dish in 10 ml of 5% FBS/DMEM with $1 \times$ Anti Anti, and grown at 37°C in 5% CO$_2$ for 72 hr. Media in the dishes was collected, filtered through a 0.22 μm pore syringe filter, and concentrated 8-fold using Amicon Ultra-4 centrifugal filter devices (3 kDa MW-cutoff) as described in the manufacturer's protocol. The cells in each

dish were detached with Versene (5.4 mM EDTA) solution, pelleted by centrifugation at 1000 RPM for 5 min at 4°C, rinsed in 10 ml of 1 × DPBS, pelleted, and re-suspended in 1.5 ml of cell lysis buffer (50 mM Tris-HCL, 150 mM NaCl, 0.5 mM EGTA-NaOH, 0.3 mM $CaCl_2$, and 0.2% w/v n-dodecyl β-d-maltoside) supplemented with 1 mM PMSF and Protease Inhibitor Cocktail (Roche). Lysates were incubated for 15 min on ice, centrifuged at 14,000 RPM for 10 min at 4°C to pellet genetic material released from nuclei. Supernatants were collected. Total [protein] was determined using the Pierce BCA Protein Assay kit (Thermo Fisher cat. # 23227).

## Capture ELISA

A high-binding, white, 96-well microtiter plate (Thermo, 7572) was coated with 100 µl/well of rabbit α-ANXA1 antibody (Proteintech, 21990–1-AP) diluted 1:500; or rabbit α-β actin antibody (Cell Signaling Technologies, 7881S) diluted 1:100 in carbonate capture buffer (6 mM $Na_2CO_3$; 44 mM $NaHCO_3$; pH 9.6). Plates were incubated overnight at 4°C. The next day, the capture antibody solution was discarded, wells were rinsed 3 × with 200 µl of wash solution (50 mM Tris, 0.14 M NaCl, 0.05% Tween 20, pH 8.0), and 200 µl of blocking solution (50 mM Tris, 0.14 M NaCl, 1% BSA, pH 8.0) was added to each well and incubated for 30 min at RT. The wells were rinsed 3 × again as described before. 100 µl of each concentrated cell-culture media (supernatant) or cell lysate sample were added. Five-fold serial dilutions were made of concentrated supernatants (neat, 1:5, 1:25, 1:125 in cell culture media) and cell lysates (0.5, 0.1, 0.02, 0.004, 0.0008 mg/mL in cell lysis buffer) in six replicates. Wells containing cell-culture media or cell lysis buffer only were made in triplicate as negative controls. The plate was incubated overnight at 4°C. The next day, samples in the wells were discarded, and the wells rinsed 3×. 100 µl of primary mouse α-ANXA1 antibody (ECM Biosciences, AM0211) diluted 1:1000; or mouse α-pan actin antibody (Cell Signaling Technologies, 7881S) diluted 1:100 in antibody diluent (50 mM Tris, 0.14 M NaCl, 1% BSA, 0.05% Tween 20, pH 8.0) was added to the appropriate wells and incubated for 1 hr at RT. Contents in the wells were again discarded and the wells rinsed 3×. 100 µL of secondary goat α-mouse IgG-HRP antibody (Cell Signaling Technologies, 7074S) diluted 1:2000 in antibody diluent was added to each well, and incubated for 1 hr at RT. Again, contents in the wells were discarded and the wells rinsed 3×. 100 µL of PICO chemiluminescent HRP substrate (Pierce) was added to each well and incubated for 5 min at RT. Chemiluminescence was measured in a Cytation five plate reader (BioTek).

# $InsP_3R$-mediated intracellular $Ca^{2+}$ release in HEKtsA201 cells expressing different levels of ANXA1

## Acute knock-down of ANXA1 expression in HEKtsA201 cells by siRNA transfection

We performed knock-down experiments by transfecting human HEKtsA201 cells with human ANXA1 siRNA. For comparison, two kinds of controls were performed: HEKtsA201 cells were either transfected with non-targeting (N-T) siRNA or not transfected at all. Four pairs of controls and experiments were done: two pairs were experiments using ANXA1 siRNA-transfected cells versus controls using N-T siRNA-transfected cells, the other two were experiments using ANXA1 siRNA-transfected cells versus controls using non-transfected cells.

Following the siRNA transfection protocol from Dharmacon (Lafayette, CO), human ANXA1 siRNA (Dharmacon cat. # M-011161-01-0005) and N-T siRNA (Dharmacon cat. # D-001206-13-05) were suspended in 250 µl of 1 × siRNA buffer (50 µl of 5 × siRNA buffer (Dharmacon cat. # B-002000-UB-100) in 200 µl RNase-free water) to a final siRNA concentration of 20 µM, aliquoted and stored at −20°C. For a pair of control and experiment, 10 ml of HEKtsA201 cell suspension with density of ~0.26 × $10^6$ cells/ml were plated into each of four 10 cm culture dishes (two for the experiment and two for the control) and incubated overnight (~18 hr) at 37°C. To transfect two 10 cm dishes of HEKtsA201 cells, 30 µl of 20 µM siRNA (ANXA1 or N-T) was thawed and diluted with 90 µl of RNase-free water. 3.9 ml of Opti-MEM (Gibco cat. # 31985–070) was added to the siRNA, mixed by gentle pipetting. The siRNA mixture was incubated for 7 min. For a non-transfected control, 4 ml of Opti-MEM was used. 160 µl of transfection reagent (Dharmacon cat. # T-2001–02) was diluted with Opti-MEM to a final volume of 8 ml, mixed by gentle pipetting. The resulting transfection solution was incubated for 7 min. 4 ml of transfection solution was added to the 4 ml of siRNA mixture (for experiment ANXA1 or control N-T siRNA transfection) or 4 ml of Opti-MEM solution (for non-

transfection control). The resulting mixtures were incubated for another 25 min. Each of the mixtures was then diluted with 32 ml of antibiotic-free complete medium (DMEM (Corning cat. # 10–013-CV) with 10% FBS (HyClone cat. # SH30071.03) to give 40 ml of transfection medium.

Culture medium was removed from each of the four dishes of HEKtsA201 cells that had been incubated overnight. 20 ml of ANXA1 siRNA transfection medium was added to each of two dishes for the experiment; and 20 ml of N-T siRNA transfection medium or non-transfection medium was added to each of two dishes for the control. The cells in transfection medium were incubated for 6 hr at 37°C. After the incubation, the transfection medium was replaced with 20 ml of regular HEKtsA201 cell medium for each of the four dishes of cells. Cells were incubated at 37°C for an additional 42 hr before being used for measurements of intracellular $Ca^{2+}$ release.

Expression levels of ANXA1 in HEKtsA201 cells were probed in Western blots using anti-ANXA1 antibody (Proteintech cat. # 21990–1-AP, 1:1000 dilution), stripped and re-probed with anti-β-actin antibody (Cell Signalling cat. # 8H10D10, 1:1000 dilution).

## Spectrofluorimetry of intracellular $Ca^{2+}$ release stimulated by extracellular agonist

The same protocol was used to perform fluorimetry for experiment and control HEKtsA201 cells. After incubation, ~10–20 × $10^6$ cells (ANXA1 siRNA transfected, N-T siRNA transfected or non-transfected) were in each 10 cm culture dish. To collect the attached cells, the HEKtsA201 medium was removed, the cells were rinsed once with 10 ml of $Ca^{2+}$-free DPBS (Corning cat. # 21–031 CM) at RT, incubated in 5 ml of Versene solution (Gibco cat. # 15040–066) for 20 min to detach, dispersed by gently pipetting the cell suspension, and the cell density was determined using a hemocytometer. Four million cells were used to quantify the expression level of ANXA1 in the HEKtsA201 cells by Western blotting. The remaining cells were centrifuged at 350 $g$ for 3 min. The supernatant was discarded, and the cell pellet was resuspended in 10 ml of Fura-2 AM dye-loading solution (1 μM Fura-2 AM dye (Molecular Probes cat. # I-1225) and 0.01% pluronic F127 (Thermo Fisher cat. # P3000MP) in HEPES-buffered saline (HBS) containing 135 mM NaCl, 5.9 mM KCl, 1.2 mM $MgCl_2$, 1.5 mM $CaCl_2$, 11 mM glucose, and 11.6 mM Na-HEPES (pH 7.4). The cell suspension was kept in darkness with gentle shaking for 20 min at RT. The dye-loaded cells were then centrifuged at 350 $g$ for 3 min. The supernatant was discarded, and the cell pellet was resuspended in 10 ml HBS. The cell suspension was again centrifuged at 350 $g$ for 3 min. The supernatant was discarded, and the cell pellet was resuspended in HBS to a cell density of 3–4 × $10^6$ cells/ml. These cells were kept in darkness. Each 10 cm dish of cells was sufficient for 4 to 7 assays. For proper control, dishes of ANXA1 siRNA-transfected cells and control cells were used alternately. Just before each assay, 0.75 ml of cell suspension was centrifuged in a 1.5 ml eppendorf tube at 400 $g$ for 3 min. The supernatant was discarded and the cells were resuspended in 0.75 ml of HBS and transferred to a cuvette containing 0.75 ml of HBS and a 3–mm magnetic stirrer. The cuvette was transferred to the temperature-controlled (37°C) experimental compartment of a multi-wavelength-excitation dual-wavelength-emission high-speed spectrofluorometer (Delta RAM, Photon Technology International). Fluorescence at 340 nm excitation/515 nm emission and 380 nm excitation/515 nm emission for the $Ca^{2+}$-bound and $Ca^{2+}$-free Fura-2 species, respectively, were measured at 5 Hz. The ratio ($R$) of the fluorescence intensity excited by 340 nm versus 380 nm light was used as relative measurement of intracellular $[Ca^{2+}]_{free}$. After a stable baseline of $R$ had been observed for 90 s, 15 μl of 10 mM carbachol solution was added to the cell suspension (final 10 μM carbachol) to stimulate sub-maximal $InsP_3R$-mediated release of $Ca^{2+}$ from the ER. $R$ was further observed for another 120 s.

## Imaging histamine-induced $Ca^{2+}$ signals in HeLa cells expressing different levels of ANXA1

HeLa cells were cultured on 15 cm tissue culture dishes in Dulbecco's Modification of Eagle's (DMEM Corning) supplemented with 10% fetal bovine serum (Hyclone) and maintained at 37°C in a humidified incubator gassed with 95% air and 5% $CO_2$. Transfection with siRNA for either human *ANXA1* siRNA or non-targeting siRNA control was performed as described for HEKtsA201 cells. ~ 3 × $10^5$ cells were seeded on each 12 mm coverslip. At 48 hr post-transfection, cells were loaded with 2 μM Fura-2 AM for 15–20 min then incubated in 2 Ca Tyrode's solution for 5–10 min at room temperature prior to imaging. Cells were imaged on a Nikon Ti system using a 20x/0.75 NA

objective for fluorescence at 340 nm excitation/515 nm emission ($Ca^{2+}$-bound Fura2), and 380 nm excitation/515 nm emission ($Ca^{2+}$-free Fura2). Coverslips were perfused with 2 Ca Tyrode's solution for 30 s, 10 or 100 $\mu$M histamine in 2 Ca Tyrode's solution for 360 s or 120 s respectively, followed by a washout with 2 Ca Tyrode's solution for 90 s. 100 ms exposure images for each wavelength were collected every 2 s. For analysis of intracellular $Ca^{2+}$-signals, cells were mass-selected using a binary mask and the Time Series Analyzer v3 plugin in ImageJ (NIH). The 340 nm/380 nm ($R_{fura2}$) fluorescence ratio for each time point was measured for every cell in the field of view and the normalized signal ($R/R_0$) was calculated by dividing $R_{fura2}$ for each time point by the mean $R_{fura2}$ during the first 30 s of each experiment. A series of logic tests were applied to data to identify signaling cells and separate those cells into oscillating and non-oscillating groups. To exclude cells that did not begin and end at baseline $Ca^{2+}$ levels, only cells with $R/R_0$ values between 1.25–0.75 during the first 30 s and $R/R_0$ values less than 1.5 during the last 30 s of the final washout were selected. To exclude non-responding cells, only cells with maximal $R/R_0$ values $\geq$ 2.0 following histamine stimulation were selected. Cells with a coefficient of variation (%CV) value greater than 15% measured from the end of the initial $Ca^{2+}$ signal to the end of the histamine stimulation were considered oscillating cells. Oscillating cells were qualitatively characterized as exhibiting either 'periodic oscillation' or 'random spiking' pattern.

## Monitoring intracellular elementary $Ca^{2+}$ release events ($Ca^{2+}$ puffs) in HEK293 cells by total internal reflection microscopy

### Cell culture and siRNA transfection

WT HEK-293 cells were cultured on plastic 75 cm tissue culture flasks in Eagle's Minimum Essential Medium (EMEM; ATCC #30–2003) supplemented with 10% fetal bovine serum (Omega Scientific #FB-11), and maintained at 37°C in a humidified incubator gassed with 95% air and 5% $CO_2$. For transfection with siRNA, cells were collected using 0.25% Trypsin-EDTA (Gibco #25200–056) and grown on 60 mm culture dishes. When cells reached 80% confluence they were incubated with 15 nM *ANXA1* siRNA or with 15 nM non-targeting CTL siRNA together with 15 $\mu$l of DharmaFECT transfection reagent (GE Healthcare) for 6 hr. In some experiments, cells were additionally incubated with 5 nM siGLO Red Transfection Indicator (Dharmacon #D-001630-02-05) to serve as a fluorescent tracer to visualize transfection efficiency. Following incubation, cells were harvested using 0.25% Trypsin-EDTA, pelleted by low speed centrifugation (~800 x g), resuspended in EMEM and distributed among poly-D-lysine coated (1 mg/ml; Sigma #P0899) 35 mm glass-bottom imaging dishes (MatTek #P35-1.5–14 C) where they were grown for 48 hr prior to imaging, at which time cells were ~50% confluent.

Immediately before imaging, cells were incubated with membrane-permeable esters of the fluorescent $Ca^{2+}$ dye Cal-520/AM (5 $\mu$M; AAT Bioquest #21130) and the caged $IP_3$ analogue ci-$IP_3$/PM [D-2,3,-O-Isopropylidene-6-O-(2-nitro-4,5 dimethoxy) benzyl-myo-Inositol 1,4,5,-trisphosphate Hexakis (propionoxymethyl) ester] (1 $\mu$M; SiChem #cag-iso-2-145-10) for 1 hr at RT in a $Ca^{2+}$-containing HEPES buffered salt solution ($Ca^{2+}$-HBSS). Cells were then washed with $Ca^{2+}$-HBSS and loaded for an additional hour with EGTA/AM (5 $\mu$M; ThermoFisher #E1219). Cal-520/AM, caged i-$IP_3$/PM, and EGTA/AM were all solubilized with DMSO/20% pluronic F127 (ThermoFisher #P3000MP). $Ca^{2+}$-HBSS contained (in mM) 135 NaCl, 5.4 KCl, 2 $CaCl_2$, 1 $MgCl_2$, 10 HEPES, and 10 glucose (pH = 7.4 at RT).

### $Ca^{2+}$ imaging

Total internal reflection fluorescence (TIRF) imaging of $Ca^{2+}$ signals was accomplished using a home-built system, based around an Olympus IX50 microscope equipped with an Olympus 60X objective (NA 1.45). Using 488 nm laser fluorescence excitation and a 510 nm long pass emission filter, fluorescence images were captured with an Evolve EMCCD camera (Photometrics), utilizing 2 $\times$ 2 pixel binning for a final field of 128 $\times$ 128 pixels (one pixel = 0.53 $\mu$m) at a rate of ~125 frames s$^{-1}$. To photo-release i-$IP_3$, UV light from a xenon arc lamp was filtered through a 350–400 nm bandpass filter and introduced by a UV-reflecting dichroic in the light path to uniformly illuminate the field of view. The amount of i-$IP_3$ released was controlled by varying the flash duration, set by an electronically controlled shutter (UniBlitz). All image data were streamed to computer memory using Metamorph v7.7 (Universal Imaging/Molecular Devices) and stored on hard disc for offline analysis.

## Image analysis

Image data in MetaMorph stk format were processed using Flika (*Ellefsen et al., 2014*), a freely available open-source image processing and analysis software written in the Python programming language (*Ellefsen et al., 2019*). Fluorescence records of 30 s and 10 s immediately following the UV flash were analyzed for the 50 ms and 150 ms flash durations, respectively; shorter records were analyzed for the 150 ms flash duration because of rapidly rising global $Ca^{2+}$ levels which hamper measurement of local $Ca^{2+}$ signals. Raw fluorescence records were first black-level subtracted and then processed to create a ratio image stack ($F/F_0$), where the fluorescence intensity of each pixel in a given frame was replaced by that intensity (F) divided by the mean resting fluorescence intensity at that pixel averaged over 500 frames prior to photo-release of i-IP$_3$ ($F_0$). Next, a custom plug-in ('detect puffs') was applied for automated detection and analysis of local $Ca^{2+}$ signals. All $Ca^{2+}$ puffs identified by the algorithm were verified by visual inspection prior to further analysis. Measurement of peak puff amplitudes ($\Delta F/F_0$) and kinetics were performed by the algorithm on a $5 \times 5$ pixel region of interest centered over the centroid of each event, and were exported to EXCEL spreadsheets for further analysis. Additional analysis and graphing was performed in Microcal Origin v6.0 (OriginLab) and Igor 6 (WaveMetrics).

## Quantification and statistical analyses

All measurements in this study: InsP$_3$R channel open probabilities ($P_o$), normalized probabilities of detection of InsP$_3$R channel activity ($P_d$), normalized maximal change of fluorescence intensity ratio ($\Delta R_{max}$) and normalized rate of change of fluorescence intensity ($1/\tau$), number of $Ca^{2+}$ puffs per second per imaging field and amplitudes of the $Ca^{2+}$ puffs ($\Delta F/F_0$), performed under various experimental conditions were compared using unpaired *t*-test. For data sets involved in multiple comparisons, *p* values were determined using the Bonferroni correction for multiple comparisons. Quantified western blot intensities from co-immunoprecipitation experiments were compared by paired *t*-test.

## Acknowledgements

This work was supported by NIH grants R37 GM 56328 (JKF), R01 GM 114042 (D-ODM) and GM 048071 (IP).

## Additional information

### Funding

| Funder | Grant reference number | Author |
|---|---|---|
| National Institutes of Health | R37GM56328 | J Kevin Foskett |
| National Institutes of Health | R01GM114042 | Don-On Daniel Mak |
| National Institutes of Health | GM048071 | Ian Parker |

The funders had no role in study design, data collection and interpretation, or the decision to submit the work for publication.

### Author contributions

Horia Vais, Conceptualization, Data curation, Supervision, Investigation, Writing - review and editing; Min Wang, Chris McKennan, Matthew Yan-lok Chan, Data curation, Formal analysis, Investigation; Karthik Mallilankaraman, Data curation, Investigation, Methodology; Riley Payne, Data curation, Formal analysis, Supervision, Investigation, Writing - review and editing; Jeffrey T Lock, Lynn A Spruce, Data curation, Formal analysis, Investigation, Methodology; Carly Fiest, Data curation, Investigation; Ian Parker, Data curation, Formal analysis, Supervision, Funding acquisition, Methodology, Writing - review and editing; Steven H Seeholzer, Conceptualization, Data curation, Formal analysis, Supervision, Methodology, Writing - original draft; J Kevin Foskett, Conceptualization, Formal analysis, Supervision, Funding acquisition, Validation, Writing - original draft, Project administration, Writing - review and editing; Don-On Daniel Mak, Conceptualization, Data curation, Formal analysis,

Supervision, Funding acquisition, Investigation, Writing - original draft, Project administration, Writing - review and editing

## Author ORCIDs
Karthik Mallilankaraman [iD] http://orcid.org/0000-0002-9492-9050
Jeffrey T Lock [iD] http://orcid.org/0000-0003-1522-3189
Carly Fiest [iD] http://orcid.org/0000-0002-1162-712X
J Kevin Foskett [iD] https://orcid.org/0000-0002-8854-0268
Don-On Daniel Mak [iD] https://orcid.org/0000-0001-7869-9382

## Decision letter and Author response
Decision letter https://doi.org/10.7554/eLife.53531.sa1
Author response https://doi.org/10.7554/eLife.53531.sa2

## Additional files

### Supplementary files
• Transparent reporting form

### Data availability
All data generated and analyzed are included in the manuscript and supporting files.

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
