## [Decision Letter]

**Acceptance summary:**

In this manuscript, Vais et al. use an elegant experimental approach to address the seemingly intractable question of whether the high Ca^2+^ concentration [Ca^2+^] in the endoplasmic reticulum (ER) lumen regulates the activity of inositol 1,4,5-trisphosphate receptors (InsP_3_Rs) independent of flow-through effects on the cytoplasmic surface of the channel.

**Decision letter after peer review:**

Thank you for submitting your article "Profound ER-luminal [Ca^2+^] regulation of InsP_3_ receptor gating mediated by an ER-luminal peripheral Ca^2+^-binding protein" for consideration by *eLife*. Your article has been reviewed by three peer reviewers, including Mark T Nelson as the Reviewing Editor and Reviewer #1, and the evaluation has been overseen by Kenton Swartz as the Senior Editor. The following individual involved in review of your submission has agreed to reveal their identity: Colin W Taylor (Reviewer #3).

The reviewers have discussed the reviews with one another and the Reviewing Editor has drafted this decision to help you prepare a revised submission.

Summary:

In this manuscript, Vais et al. use an elegant experimental approach to address the seemingly intractable question of whether the high Ca^2+^ concentration [Ca^2+^] in the endoplasmic reticulum (ER) lumen regulates the activity of inositol 1,4,5-trisphosphate receptors (InsP_3_Rs) independent of flow-through effects on the cytoplasmic surface of the channel. The authors show that, under conditions in which [Ca^2+^] is controlled so as to produce a physiologically high concentration in the lumen (300 nM) and appropriately low concentration in cytoplasm-equivalent spaces (70 nM), InsP_3_R open probability in the presence of cytosolic IP_3_ was profoundly reduced compared with that observed under symmetrically low [Ca^2+^] (70 nM in ER and cyto) conditions. The absence of an inhibitory effect upon elevating [Ca^2+^] in the ER-equivalent space to 300 μM while recording in the lumen-out configuration suggested the presence of a loosely associated inhibitory lumen protein, which they termed peripheral luminal-protein (PLP). After demonstrating that this PLP-mediated inhibitory phenomenon was generalizable to other cell types expressing different repertoires of InsP_3_Rs, they further established that a region of the L2 loop in common among all InsP_3_R subtype was responsible for PLP binding and that an excess of isolated peptide with this sequence was capable of blocking the inhibitory effect of PLP. Using this peptide to pull down proteins from soluble protein fractions of bovine hepatocyte ER microsomes in conjunction with mass spectrometry, they identified PLP as a member of the annexin (ANX) family of Ca^2+^-binding proteins. Subsequent experiments performed using recombinant ANXA1, ANXA6 and ANXa11, all of which were detected among ER proteins, and provide evidence that ANXA1 as the isoform responsible for the inhibitory effect and showed that endogenous ANXA1 was capable of inhibiting InsP_3_R-mediated Ca^2+^ release in intact cells.

Essential revisions:

1) Readers without a background in electrophysiology may be confused by the off-hand allusion to recording K^+^ currents; given the familiar role of InsP_3_Rs in mediating Ca^2+^ efflux, these readers would be expecting Ca^2+^ to be monitored directly. This same demographic will also likely have difficulty orienting to the techniques used to create lumen-out and cytoplasm-out patch configurations. Because such "naïve" readers, which include molecular and cell biologists, and many others, are among those most likely to be interested in this work, manoeuvres used to record K^+^ conductance (instead of Ca^2+^ flux) and create lumen-out and cytoplasm-out patches should be described in plain English (and depicted schematically, if possible).

2) Although DT40-KO-r-InsP_3_R-3 cells used for the bulk of the experiments are well described in the Materials and methods, it would be helpful if they were more clearly described when first introduced in Results (e.g., "…isolated from chicken DT40-KO-r-InsP_3_R-3 cells, in which all endogenous InsP_3_Rs have been stably deleted and replaced with rat type-3 InsP_3_R channels.").

3) The reported effect of the luminal modulation/inhibition of InsP_3_Rs is profound (e.g. strong effect on *P*_o_) and universal (demonstrated for all 3 types of InsP_3_Rs). An impressive combination of advanced electrophysiological techniques and molecular biology approaches allowed the authors to convincingly demonstrate that Ca^2+^ concentration in the ER lumen has strong modulatory effect on the properties of InsP_3_Rs. The authors also provided compelling evidence for the involvement of a luminal Ca^2+^-binding protein in this modulation. There was a consensus amongst the reviewers that this phenomenon is novel, important and well-documented, and could stand on its own. It was felt, however, that the role of ANXA-1 is less strong. The major criticism relates to the suggestion that ANXA1 mediates endogenous inhibition by luminal Ca^2+^. The evidence (Figure 5) that ANXA1 is expressed in the 'ER lumen' is not clear. There is evidence that ANXA1 can confer an effect of luminal [Ca^2+^] (Figure 6-8), but that is not in itself evidence that it is the endogenous mediator. While the relatively modest effects of siRNA-mediated knockdown of ANXA1 on Ca^2+^ signals in intact cells lend support to its role, there is no evidence that these effects are related to luminal [Ca^2+^]. Immunostaining for AnxA1 looks like staining for nuclear lamins rather than the expected ER distribution. It would be important to verify the presence of AnxA1 in the ER lumen (e.g. by EM or super-resolution microscopy) and perhaps discuss/explain the retention of AnxA1 in this perinuclear compartment (and is consequences for cell physiology).

The authors should bolster their evidence for ANXA1 (within 2 months) or consider it for a follow-up paper.

4) The reviewers agree that phenomenon is specific for the perinuclear ER, but it is unclear if it applies to other ER compartments. The authors should provide evidence for this mechanism in other ER compartments, which is tough, or limit their conclusion to the perinuclear ER and discuss its general relevance.

5) Figure 2: the argument that the effects of cytosolic ATP (via luminal [Ca^2+^] are not due to 'feed-through' to a cytosolic site rest on the argument that diBrBAPTA should buffer such effects (subsection “Novel Regulation by [Ca^2+^]_ER_ of InsP_3_R Single-Channel Activity”, second paragraph). This may be true, but the argument would be stronger if the same effects of ATP were also shown at holding potentials where any Ca^2+^ flux would be directed into the lumen.

6) Figure 2H (and subsection “Novel Regulation by [Ca^2+^]_ER_ of InsP_3_R Single-Channel Activity”, third paragraph) suggests a substantial increase in the frequency of gating and a probable decrease in *P*_o_ in lumen-out recordings when [Ca^2+^]lumen was increased to 300μM, whereas the summary data (J) and subsequent conclusions (the need for an accessory protein) suggest no change in *P*_o_. Please clarify.

7) Figure 3H: it would be preferable to see this figure presented with the actual recorded *P*_o_, rather than 'normalizing' data across different sets of experiments.

8) Figure 4J: the results shown do not establish that the increased *P*_o_ achieved in the presence of pL2 is because it reverses the inhibition by luminal Ca^2+^. It needs to be shown that pL2 does not increase *P*_o_ under conditions with reduced luminal [Ca^2+^].

---

## [Author Response]

Essential revisions:1) Readers without a background in electrophysiology may be confused by the off-hand allusion to recording K^+^ currents; given the familiar role of InsP_3_Rs in mediating Ca^2+^ efflux, these readers would be expecting Ca^2+^ to be monitored directly. This same demographic will also likely have difficulty orienting to the techniques used to create lumen-out and cytoplasm-out patch configurations. Because such "naïve" readers, which include molecular and cell biologists, and many others, are among those most likely to be interested in this work, manoeuvres used to record K^+^ conductance (instead of Ca^2+^ flux) and create lumen-out and cytoplasm-out patches should be described in plain English (and depicted schematically, if possible).

We have now modified the section on nuclear patch-clamp electrophysiology methods to provide better descriptions of the methods we used to study the InsP_3_R channel activity (Introduction). Schematic diagrams showing different nuclear patch-clamp configurations (on-nucleus, luminal-side-out and cytoplasmic-side-out) have now been added in a new Figure 1. So all subsequent figure numbers are increased by one. We have also now clarified why the electrical currents through open InsP_3_R channels observed in our nuclear patch-clamp experiments were carried by K^+^ and not Ca^2+^.

2) Although DT40-KO-r-InsP_3_R-3 cells used for the bulk of the experiments are well described in the Materials and methods, it would be helpful if they were more clearly described when first introduced in Results (e.g., "…isolated from chicken DT40-KO-r-InsP_3_R-3 cells, in which all endogenous InsP_3_Rs have been stably deleted and replaced with rat type-3 InsP_3_R channels.").

We have now added a description of the DT40-KO-r-InsP_3_R-3 cell line used in our experiments in the main text (subsection “Novel Regulation by [Ca^2+^]_ER_ of InsP_3_R Single-Channel Activity”, second paragraph).

3) The reported effect of the luminal modulation/inhibition of InsP_3_Rs is profound (e.g. strong effect on P_o_) and universal (demonstrated for all 3 types of InsP_3_Rs). An impressive combination of advanced electrophysiological techniques and molecular biology approaches allowed the authors to convincingly demonstrate that Ca^2+^ concentration in the ER lumen has strong modulatory effect on the properties of InsP_3_Rs. The authors also provided compelling evidence for the involvement of a luminal Ca^2+^-binding protein in this modulation. There was a consensus amongst the reviewers that this phenomenon is novel, important and well-documented, and could stand on its own. It was felt, however, that the role of ANXA-1 is less strong. The major criticism relates to the suggestion that ANXA1 mediates endogenous inhibition by luminal Ca^2+^. The evidence (Figure 5) that ANXA1 is expressed in the 'ER lumen' is not clear. There is evidence that ANXA1 can confer an effect of luminal [Ca^2+^] (Figure 6-8), but that is not in itself evidence that it is the endogenous mediator. While the relatively modest effects of siRNA-mediated knockdown of ANXA1 on Ca^2+^ signals in intact cells lend support to its role, there is no evidence that these effects are related to luminal [Ca^2+^]. Immunostaining for AnxA1 looks like staining for nuclear lamins rather than the expected ER distribution. It would be important to verify the presence of AnxA1 in the ER lumen (e.g. by EM or super-resolution microscopy) and perhaps discuss/explain the retention of AnxA1 in this perinuclear compartment (and is consequences for cell physiology).The authors should bolster their evidence for ANXA1 (within 2 months) or consider it for a follow-up paper.

We described in the Discussion of the original version of the manuscript two important caveats regarding the conclusion that ANXA1 was *the* luminal protein: our inability to immunoprecipitate it with the InsP_3_R, and to convincingly localize it to the ER beyond the nuclear envelope. On the other hand, a substantial body of evidence presented here demonstrates that ANXA1 has all of the properties of the peripheral membrane-associated Ca^2+^ binding protein (PLP) that, as the reviewer notes, we have convincingly demonstrated exists We discuss this in a separate Results section of the manuscript. Accordingly, we have decided to leave the ANXA1 data in the manuscript while providing much stronger caveats and reservations regarding our ability to conclude that ANXA1 is the ubiquitous PLP that regulates the InsP_3_R. In the revised manuscript, we do the following. We have re-written sentences to avoid conclusions that ANXA1 is the PLP, by including caveat words such as “likely”, or simply not mentioning ANXA1 but instead referring to PLP. In addition, we have more strongly emphasized the section in the Discussion where we point to the IP and localization failures. We now point that all of our patch-clamp studies were performed on the nuclear envelope, the same compartment where we were able to localize ANXA1 to the nuclear envelope lumen, suggesting that our observation regarding ANXA1 might be relevant for this compartment specifically. And we now add a new section at the end of the manuscript that points out limitations of the study, and suggest experiments that could be done in future studies to more definitively identify the PLP. Together, we feel that the reader now will have the strong sense that ANX1 could be the protein, but establishing it as *the* PLP remains to be determined.

4) The reviewers agree that phenomenon is specific for the perinuclear ER, but it is unclear if it applies to other ER compartments. The authors should provide evidence for this mechanism in other ER compartments, which is tough, or limit their conclusion to the perinuclear ER and discuss its general relevance.

Please see previous response. Because of our inability to localize ANXA1 to the entire ER, we now suggest in the limitations of the current study that the phenomena we document could possibly be applicable specifically to the perinuclear ER.

5) Figure 2: the argument that the effects of cytosolic ATP (via luminal [Ca^2+^] are not due to 'feed-through' to a cytosolic site rest on the argument that diBrBAPTA should buffer such effects (subsection “Novel Regulation by [Ca^2+^]_ER_ of InsP_3_R Single-Channel Activity”, second paragraph). This may be true, but the argument would be stronger if the same effects of ATP were also shown at holding potentials where any Ca^2+^ flux would be directed into the lumen.

To clarify the experiments shown in Figure 3 (previously Figure 2), we have now added three schematic diagrams (Figures 3A, C and E). We already previously performed the kinds of experiments that the reviewer mentioned, in which the high [Ca^2+^]_ER_ is prevented from generating a feed-through effect by raising the holding potential high enough to oppose the Ca^2+^ concentration gradient across the ER membrane, in: J. Gen. Physiol. 140: 691-716 (2012) Figure 3 vs. Figure 8. We note that even using voltage to impede Ca^2+^ flux, a high concentration of Ca^2+^ chelator on the cytoplasmic side of the channel is still needed to buffer the Ca^2+^ passing through the channel. That is, the electrical potential needed to abrogate the feed-through effect using only a low concentration of chelator (0.5 mM) is much higher than the membrane patch at the pipette tip can tolerate.

6) Figure 2H (and subsection “Novel Regulation by [Ca^2+^]_ER_ of InsP_3_R Single-Channel Activity”, third paragraph) suggests a substantial increase in the frequency of gating and a probable decrease in P_o_ in lumen-out recordings when [Ca^2+^]lumen was increased to 300μM, whereas the summary data (J) and subsequent conclusions (the need for an accessory protein) suggest no change in P_o_. Please clarify.

In Figure 3K (previously Figure 2H), only a total 1.2 s of the “typical” current trace was shown to give the readers some idea of the gating of the InsP_3_R channels under various ligand conditions. In general, the gating of the channels is stochastic. The current traces we used in our studies were significantly longer (tens of seconds to tens of minutes). Even so, the *P*_o_ of individual channel records varied substantially, as shown in Figures 3 L-M (previously Figures 2 I-J), so significant number of current traces (tabulated in the graphs) were collected to obtain meaningful statistics for our experiments.

As for the current trace in Figure 3K (previously Figure 2H) that the reviewer pointed out, it is obvious that the frequency of channel closings (seen as upward current spikes) in 70 nM [Ca^2+^]_ER_ before the perfusion solution switch was significantly lower than that in 300μM [Ca^2+^]_ER_ after perfusion switch. However, it should be noted that most of the channel closings in both 70 nM and 300 μM [Ca^2+^]_ER_ had very short durations, so the mean *P*_o_ of the channel recorded in 70 nM [Ca^2+^]_ER_ for the ~ 0.5 s before the solution switch is *not* substantially higher than the mean *P*_o_ recorded in 300μM for the ~ 0.5 s after the solution switch, as the reviewer suggested.

7) Figure 3H: it would be preferable to see this figure presented with the actual recorded P_o_, rather than 'normalizing' data across different sets of experiments.

We agree with the reviewer. We now show the actual recorded InsP_3_R channel *P*_o_ of endogenous WT DT40, PC12 and N2a cells under optimal [Ca^2+^]_i_ and maximal [InsP_3_] in the absence (Figure 4G, previously Figure 3G) and presence (Figure 4H, new figure) of 1 mM MgATP in the bath solution. We have also included the normalized channel *P*_o_ for the various InsP_3_R channels in +/– MgATP to show how the relative change in *P*_o_ of those channels is remarkably similar (in Figure 4I, previously Figure 3H), despite the large absolute difference among the *P*_o_ of those channels. We rearranged the averaged data in Figure 4I to avoid giving the impression that different data sets were compared.

8) Figure 4J: the results shown do not establish that the increased P_o_ achieved in the presence of pL2 is because it reverses the inhibition by luminal Ca^2+^. It needs to be shown that pL2 does not increase P_o_ under conditions with reduced luminal [Ca^2+^].

*P*_o_ is already very high under conditions of reduced luminal [Ca^2+^]. We could change the conditions, for example use lower [InsP_3_] to observe channels with a low *P*_o_ in conditions of reduced luminal [Ca^2+^]. Preliminary data from such an experiment suggest that the peptide is not stimulatory on its own. However, the number of experiments is limited due to complete shutdown of our research efforts by the COVID-19 pandemic. We point out in the new “limitations of the current study” the absence of this control experiment.